# K27-linked ubiquitination of BRAF by ITCH engages cytokine response to maintain MEK-ERK signaling

Qing Yin[1], Tao Han[1], Bin Fang [2], Guolin Zhang[3], Chao Zhang[4], Evan R. Roberts[1], Victoria Izumi[2], Mengmeng Zheng[5], Shulong Jiang [1,6], Xiu Yin[1,6], Minjung Kim[1,7], Jianfeng Cai[5], Eric B. Haura[3], John M. Koomen [1,2], Keiran S.M. Smalley[4,8] & Lixin Wan[1,8]

BRAF plays an indispensable role in activating the MEK/ERK pathway to drive tumorigenesis. Receptor tyrosine kinase and RAS-mediated BRAF activation have been extensively characterized, however, it remains undefined how BRAF function is fine-tuned by stimuli other than growth factors. Here, we report that in response to proinflammatory cytokines, BRAF is subjected to lysine 27-linked poly-ubiquitination in melanoma cells by the ITCH ubiquitin E3 ligase. Lysine 27-linked ubiquitination of BRAF recruits PP2A to antagonize the S365 phosphorylation and disrupts the inhibitory interaction with 14–3–3, leading to sustained BRAF activation and subsequent elevation of the MEK/ERK signaling. Physiologically, proinflammatory cytokines activate ITCH to maintain BRAF activity and to promote proliferation and invasion of melanoma cells, whereas the ubiquitination-deficient BRAF mutant displays compromised kinase activity and reduced tumorigenicity. Collectively, our study reveals a pivotal role for ITCH-mediated BRAF ubiquitination in coordinating the signals between cytokines and the MAPK pathway activation in melanoma cells.

[1] Department of Molecular Oncology, H. Lee Moffitt Cancer Center and Research Institute, Tampa, FL 33612, USA. [2] Proteomics and Metabolomics Core, H. Lee Moffitt Cancer Center and Research Institute, Tampa, FL 33612, USA. [3] Department of Thoracic Oncology, H. Lee Moffitt Cancer Center and Research Institute, Tampa, FL 33612, USA. [4] Department of Tumor Biology, H. Lee Moffitt Cancer Center and Research Institute, Tampa, FL 33612, USA. [5] Department of Chemistry, University of South Florida, Tampa, FL 33620, USA. [6] Department of Oncology, Jining First People's Hospital, Jining, Shandong 272111, P.R. China. [7] Department of Cell Biology, Microbiology, and Molecular Biology, University of South Florida, Tampa, FL 33620, USA. [8] Department of Cutaneous Oncology, H. Lee Moffitt Cancer Center and Research Institute, Tampa, FL 33612, USA. Correspondence and requests for materials should be addressed to L.W. (email: lixin.wan@moffitt.org)

The RAF family protein kinases consist of the ARAF, BRAF, and CRAF isoforms, which play a central role in coordinating the MEK/ERK oncogenic signaling[1]. Allosteric regulations, including phosphorylation and dimerization, govern RAF activation in response to growth factor stimulation[2]. However, how BRAF is activated in response to stimuli other than growth factors remains less understood. The tumor microenvironment (TME) has emerged as a key factor to drive tumor initiation and metastasis[3]. One of the routes through which TME influences tumor cells is by secreting inflammatory cytokines and growth factors[4]. For instance, tumor-associated macrophages have been shown to exhibit pro-tumorigenic functions to support breast tumor[5] and melanoma[6] progression. The oncogenic function of macrophages and other tumor-infiltrating immune cells has been highlighted to confer BRAF inhibition resistance in melanoma cells[6–9]. In addition to its canonical function to activate the NF-κB signaling, inflammatory cytokines, such as tumor necrosis factor α (TNFα), have been repeatedly observed to induce rapid and sustained activation of the MEK/ERK signaling pathway[10–12]. Although the increase of c-FLIP (cellular FLICE-like inhibitory protein) transcription upon NF-κB activation has been shown to regulate CRAF function in neuronal cells[11], the mechanism that leads to the immediate activation of ERK upon cytokine stimulation remains unclear.

In addition to proteasome-dependent protein degradation, protein ubiquitination has been underscored as an indispensable signaling mechanism in NF-κB pathway activation, DNA damage response, and endocytosis[13]. The versatile function of protein ubiquitination is determined, by which one of the seven lysines (K6, K11, K27, K29, K33, K48, and K63) or the N-terminal methionine (M1) on the ubiquitin molecule is used to form a polyubiquitin chain[14]. Each different ubiquitin chain linkage adopts a distinct overall conformation, and thus could be recognized by different readers to mediate different cellular functions[15]. For instance, K48- and K11-linked ubiquitin chains have been well characterized to trigger 26S proteasome-mediated proteolysis during cell-cycle progression[16], while K63- and M1-linked ubiquitin chains are assembled upon NF-κB activation during immune response[17]. However, the functions of other ubiquitin linkages are less explored.

Our findings demonstrate that in melanoma cells, upon cytokine stimulation, BRAF is subjected to nondegradable ubiquitination, primarily via the K27-linked polyubiquitin chains. This atypical ubiquitination is catalyzed by the ITCH E3 ligase in response to c-Jun-N-terminal kinase (JNK)-mediated phosphorylation. K27-linked ubiquitination of BRAF abolishes 14–3-3-mediated suppression of BRAF kinase activity, leading to sustained BRAF/MEK/ERK signaling, which contributes to the pro-tumoral cytokine-facilitated survival of melanoma cells.

## Results

**BRAF is ubiquitinated by ITCH via the K27-linkage**. We have recently reported that FZR1 promotes BRAF ubiquitination and subsequent proteolysis in normal melanocytes, whereas this process is inhibited in melanoma cells due to inhibitory phosphorylations by ERK and CDK kinases at the N terminus of FZR1[18]. Although BRAF is stabilized in melanoma cells (Supplementary Fig. 1a), polyubiquitination of endogenous BRAF was observed in both BRAF^WT- and BRAF^V600E-expressing melanoma cells (Supplementary Fig. 1b). These findings suggest that BRAF is modified by polyubiquitin chains, but not proteolytically degraded by the proteasome. To determine the polyubiquitin chain linkage assembled on the BRAF protein, BRAF was co-transfected with a panel of ubiquitin K-only constructs, in which only the indicated lysine is encoded while other six lysine residues

were mutated to arginine (Fig. 1a). Intriguingly, BRAF ubiquitination could only be carried out using WT-ubiquitin (UB) and K27-UB, and to a lesser extent, K29-UB (Fig. 1a), indicating that BRAF is mainly polyubiquitinated via the K27-linkage.

Numerous ubiquitin E3 ligases have been reported to catalyze K27-linked polyubiquitination[19–25]. Of note, HECT-type ubiquitin E3 ligases, including ITCH[24], NEDD4[23], and HACE1[22], were suggested to promote K27-linked polyubiquitination. ITCH and NEDD4 belong to the NEDD4 E3 ligase family[26]. As shown in Fig. 1b, among the five NEDD4 family E3 ligases that bind to BRAF (Supplementary Fig. 1c), only ITCH targeted BRAF for polyubiquitination. Furthermore, recombinant purified GST-ITCH promoted the polyubiquitination of immunopurified BRAF proteins in vitro (Fig. 1c). Moreover, mutation of the catalytic active cysteine residue (C832S) in ITCH abrogated ITCH-mediated BRAF polyubiquitination both in vitro (Fig. 1d; Supplementary Fig. 1d) and in cells (Supplementary Fig. 1e).

We next examined if ITCH-mediated BRAF polyubiquitination is mainly through the K27-linkage. We found that ITCH promoted both K27- and K29-, but not K48-linked, ubiquitination of BRAF (Fig. 1e). Notably, although both K27- and K29-only ubiquitin molecules could be utilized by ITCH to ubiquitinate BRAF (Fig. 1e), compared with WT- and K29R-UB, K27R-UB failed to fuel the ITCH-mediated BRAF polyubiquitination (Fig. 1f). Consistently, catalytically inactive CS-ITCH failed to promote BRAF K27-linked polyubiquitination in cells (Fig. 1g). Moreover, ITCH promoted polyubiquitination of BRAF with WT and K27 only (Fig. 1h), but not K27R-UB (Fig. 1i) in vitro. In order to determine if the 26S proteasome affects ITCH-mediated BRAF ubiquitination, in a separate experiment without the 26S proteasome inhibitor MG132, we found that ITCH efficiently assembled polyubiquitin chains on BRAF using either WT- or K27-UB (Supplementary Fig. 1f). This finding further supports a non-proteolytic role of ITCH-mediated BRAF polyubiquitination.

Similar to other NEDD4 family E3 ligases, ITCH has been shown as a versatile ubiquitin E3 ligase that promotes K48-, K29-, K33-, and K63-linked polyubiquitination on different substrates[26–29]. By utilizing a K27-linkage-specific antibody, we compared the linkage type of polyubiquitinated endogenous BRAF and c-Jun, an ITCH ubiquitin substrate that was mainly targeted via the proteolytic K48-linkage. As shown in Supplementary Fig. 1g, in contrast to the K48-linked ubiquitin chains formed on c-Jun, endogenous BRAF is mainly modified with K27-linked ubiquitin chains. Furthermore, mass spectrometry analysis using purified in vitro polyubiquitinated BRAF revealed that BRAF was ubiquitinated via the K27-linkage, as evidenced by the presence of the K27-ε-GG ubiquitin peptide (Supplementary Fig. 2a).

To pinpoint the lysine residues that are modified by ubiquitin, ubiquitinated BRAF proteins were immunopurified from 293T cells for mass spectrometry analyses. K164, K473, K570, and K698 emerged as the lysines that were ubiquitinated in cells (Fig. 1j; Supplementary Fig. 2b–f). Importantly, mutation of these lysine residues to arginine led to reduced BRAF ubiquitination, both in cells (Fig. 1k) and in vitro (Fig. 1l). In addition to the four lysine residues identified by mass spectrometry, we found that K700, but not K699, could also be ubiquitinated by ITCH (Fig. 1k, l; Supplementary Fig. 2g). These lysine residues are evolutionarily conserved in vertebrates (Supplementary Fig. 2h), suggesting a common regulatory mechanism for RAF proteins. Based on these results, we generated a BRAF mutant containing the five validated KR mutations (named 5KR, Fig. 1j) as the ubiquitination-deficient mutant that abolishes ITCH-mediated ubiquitination to WT-BRAF (Fig. 1k, l). Collectively, our results demonstrate that BRAF is ubiquitinated via the atypical K27-linkage by the ITCH ubiquitin E3 ligase.

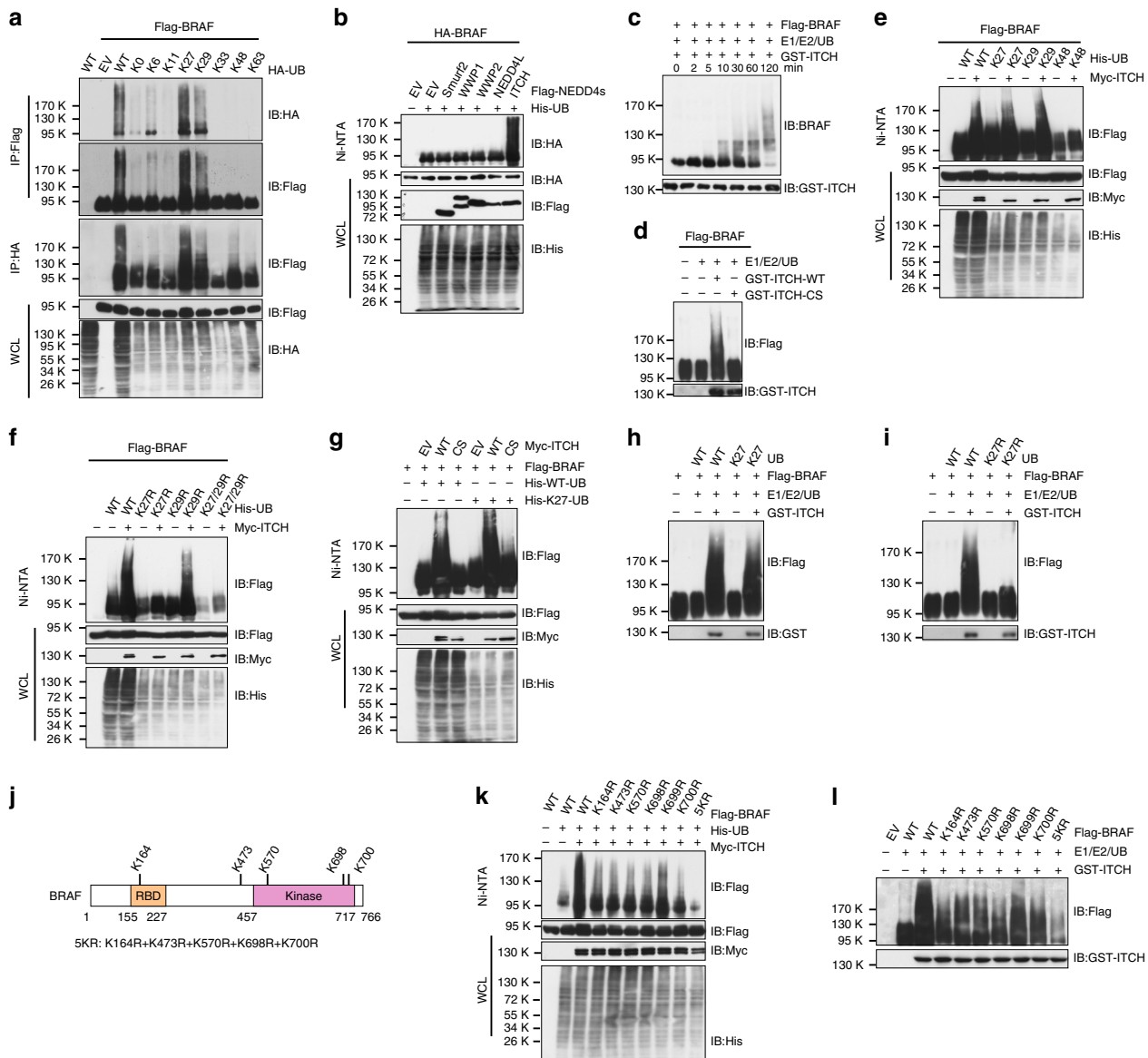

**Fig. 1** ITCH promotes K27-linked polyubiquitination of BRAF. **a** BRAF was ubiquitinated via the K27 and K29 linkages. Immunoblot (IB) analysis of whole-cell lysates (WCL) and immunoprecipitates (IP) derived from 293T cells transfected with Flag-BRAF and the indicated HA-ubiquitin (UB) constructs. **b** ITCH promoted BRAF ubiquitination in cells. IB analysis of WCL and Ni-NTA (Ni-nitrilotriacetic acid) affinity precipitates derived from 293T cells transfected with HA-BRAF, His-UB, and the indicated Flag-tagged NEDD4 family constructs. **c** ITCH promoted BRAF ubiquitination in vitro. Flag-BRAF protein immunopurified from 293T cells was incubated with GST-ITCH proteins, as well as E1, E2, and ubiquitin proteins at 30 °C for the indicated time period before SDS-PAGE and IB analyses. **d** WT- but not C832S(CS)-ITCH promoted BRAF ubiquitination in vitro. **e** BRAF was ubiquitinated via the K27- and K29 linkages. IB analysis of WCL and Ni-NTA affinity precipitates derived from 293T cells transfected with Flag-BRAF and the indicated Myc-ITCH, WT, or K-only His-UB constructs. **f** K27R-ubiquitin failed to facilitate BRAF ubiquitination. IB analysis of WCL and Ni-NTA affinity precipitates derived from 293T cells transfected with Flag-BRAF and the indicated Myc-ITCH, WT, or KR His-UB constructs. **g** BRAF was polyubiquitinated via the K27-linkage. IB analysis of WCL and Ni-NTA affinity precipitates derived from 293T cells transfected with Flag-BRAF and the indicated Myc-ITCH, His-WT-UB, or His-K27-UB constructs. **h** ITCH promoted BRAF K27-linked polyubiquitination in vitro. **i** K27R-UB failed to facilitate ITCH-mediated BRAF polyubiquitination in vitro. **j** A schematic illustration of the lysine residues identified as the BRAF ubiquitinated sites. **k** BRAF-KR mutants displayed compromised ubiquitination in cells. IB analysis of WCL and Ni-NTA affinity precipitates derived from 293T cells transfected with the indicated Flag-BRAF, Myc-ITCH, and His-UB constructs. **l** BRAF-KR mutants displayed compromised ubiquitination in vitro. Various Flag-BRAF WT and KR mutants immunopurified from 293T cells were incubated with bacterially purified GST-ITCH and the E1, E2, and ubiquitin proteins as indicated. The reaction was performed at 30 °C for 60 min and followed by SDS-PAGE and IB analyses

**ITCH interacts with BRAF through the kinase domain.** ARAF and CRAF are homologs of BRAF, which also modulate the MEK/ERK signaling[1]. We found that in addition to BRAF, ITCH interacted with (Fig. 2a) and promoted the ubiquitination of all three RAF isoforms (Fig. 2b), which is likely due to the conservation of the identified lysine residues among RAF proteins

(Supplementary Fig. 3a). Since BRAF is the predominant driver for MAPK activation in most melanoma cells, we chose to focus on elucidating how ubiquitination of BRAF by ITCH dictates the BRAF/MEK/ERK signaling cascade in the melanoma setting. To this end, we demonstrated that endogenous ITCH interacted with BRAF in both BRAF$^{WT}$-expressing WM1346 (Fig. 2c) and

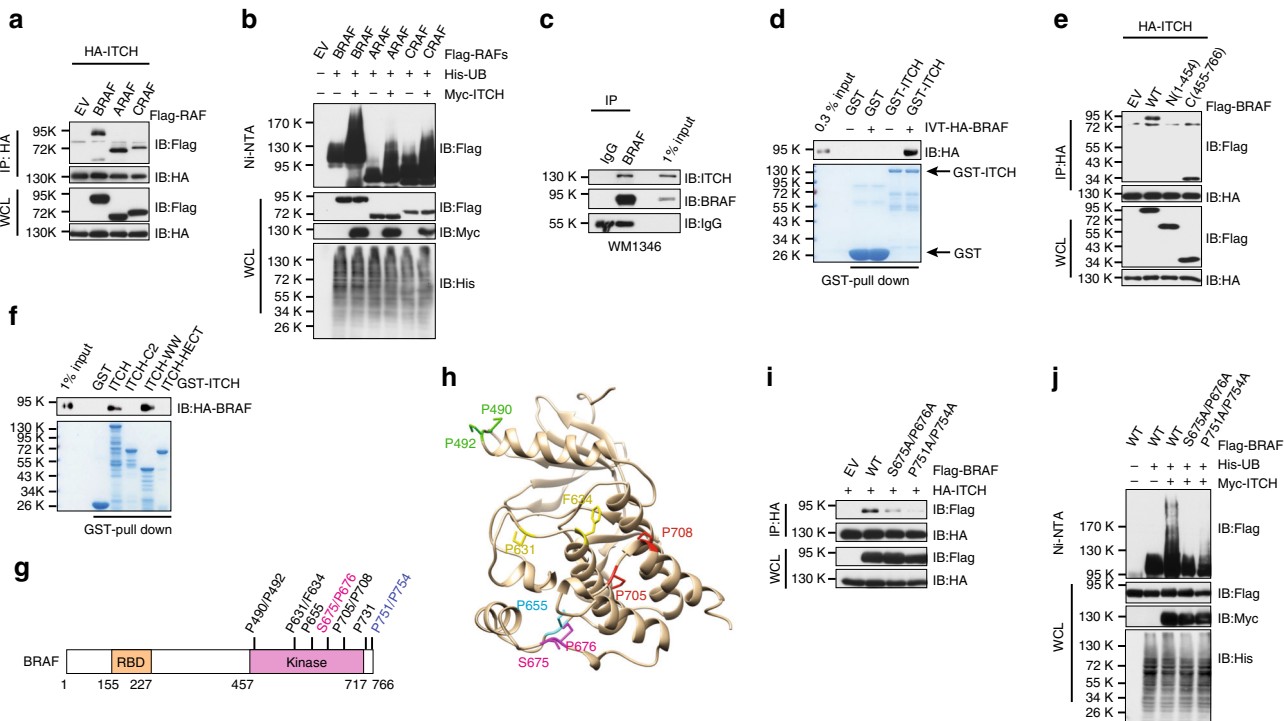

**Fig. 2** BRAF interacts with ITCH through its kinase domain. **a** ITCH bound to RAF isoforms. Immunoblot (IB) analysis of whole-cell lysates (WCL) and immunoprecipitates (IP) derived from 293T cells transfected with HA-ITCH together with the indicated Flag-RAF constructs. **b** ITCH promoted ubiquitination of RAF proteins in cells. IB analysis of WCL and Ni-NTA (Ni-nitrilotriacetic acid) affinity precipitates derived from 293T cells transfected with the indicated Myc-ITCH, His-UB, and Flag-RAF constructs. **c** Endogenous BRAF bound to ITCH. IB analysis of WCL and anti-BRAF IP derived from WM1346 cells. **d** In vitro transcribed and translated HA-BRAF (IVT-HA-BRAF) bound to purified recombinant GST-ITCH. **e** ITCH bound to the BRAF kinase domain. IB analysis of WCL and IP derived from 293T cells transfected with HA-ITCH together with the indicated Flag-BRAF constructs. **f** BRAF bound to the WW domain of ITCH in vitro. In vitro transcribed and translated HA-BRAF was incubated with the indicated recombinant GST proteins before washing and being resolved by SDS-PAGE. **g** A schematic illustration of the proline-rich motifs in the BRAF kinase domain that might mediate ITCH–BRAF interaction. **h** A structural illustration of the putative spatial location of the candidate ITCH-binding motifs in a reported BRAF crystal structure[58] (PDB ID: 1UWH). **i** Mutation of ITCH-binding motifs abrogated ITCH–BRAF interaction in cells. IB analysis of WCL and IP derived from 293T cells transfected with HA-ITCH together with the indicated Flag-BRAF constructs. **j** Mutation of ITCH-binding motifs abolished ITCH-mediated BRAF ubiquitination in cells. IB analysis of WCL and Ni-NTA affinity precipitates derived from 293T cells transfected with the indicated Myc-ITCH, His-UB, and Flag-BRAF constructs

BRAF$^{V600E}$-expressing A375 melanoma cells (Supplementary Fig. 3b). Furthermore, both BRAF$^{WT}$ and BRAF$^{V600E}$ bound to ITCH in the ectopic expression condition (Supplementary Fig. 3c). These findings were consistent with our results that ITCH catalyzes the ubiquitination of both BRAF$^{WT}$ and BRAF$^{V600E}$ (Supplementary Fig. 1e). Moreover, BRAF proteins bound to purified recombinant GST-ITCH (Fig. 2d; Supplementary Fig. 3d) in vitro, supporting a direct interaction between BRAF and ITCH molecules.

Next, we sought to determine the amino acids on the BRAF protein that mediate its interaction with ITCH. We found that the C-terminal kinase domain, but not the N-terminal region of BRAF bound to ITCH (Fig. 2e). NEDD4 family ubiquitin E3 ligases, including ITCH, interact with its substrates via their WW domains[26]. Indeed, we found that ITCH utilized its WW domain to bind BRAF (Fig. 2f). In addition to the PPxY motif[26], NEDD4 family E3 ligases recruit substrates via other proline-rich motifs, such as PPLP, pSP, or pTP motifs[30]. Numerous non-PPxY proline-rich motifs were identified within the kinase domain of BRAF (Fig. 2g, h). Seven candidate proline-rich motifs were chosen based on either their surface localization (P490/P492, P631/F634, P655, S675/P676, and P705/P708) or their existence in a disordered C-terminal region (P731, P751/P754) (Fig. 2g, h). Notably, mutating either the S675/P676 or the P751/P754 motif, but not other proline-rich motifs, attenuated the binding of BRAF to ITCH (Fig. 2i; Supplementary Fig. 3e, f). As a consequence,

both S675A/P676A- and P751A/P754A-BRAF were insensitive to ITCH-mediated polyubiquitination in cells (Fig. 2j) and in vitro (Supplementary Fig. 3g). Examination of BRAF protein sequence in various vertebrate species revealed that the ITCH-binding motifs are largely conserved (Supplementary Fig. 3h). Interestingly, both S675/P676 and P751/P754 motifs are conserved in CRAF and partly conserved in ARAF (Supplementary Fig. 3i).

Most of the identified ubiquitinated lysine residues are located within the BRAF kinase domain (Fig. 1j); to exclude the possibility that the 5KR mutant interferes with BRAF–ITCH interaction, we compared the binding of ITCH to WT-BRAF and 5KR-BRAF. As shown in Supplementary Fig. 3j, mutation of the five lysine residues in BRAF did not affect its association with ITCH. Taken together, our results pinpointed two proline-rich motifs in the BRAF kinase domain that mediates the ITCH–BRAF interaction to facilitate the K27-linked polyubiquitination of BRAF.

## ITCH depletion attenuates the MEK/ERK signaling and cell growth. Given the central role of BRAF in activating the downstream MEK/ERK pathway, we next sought to investigate if ITCH is an important upstream regulator for this signaling cascade. Compared with WT-mouse embryonic fibroblasts (MEFs), *Itch*-deleted MEFs displayed reduced MEK/ERK activity (Fig. 3a), suggesting a positive regulation of BRAF function by ITCH.

Furthermore, when treated with epidermal growth factor (EGF) or platelet-derived growth factor (PDGF), MEK/ERK activation was compromised in *Itch*-deleted MEFs (Supplementary Fig. 4a, b). Proinflammatory cytokines, such as TNFα, activate ITCH through JNK-mediated phosphorylation[31]. We found that TNFα treatment triggered rapid activation of MEK/ERK in MEFs, while deletion of *Itch* blunted the response of p-MEK/p-ERK signals upon TNFα stimuli (Fig. 3b), suggesting that cytokine-induced MEK/ERK activation is at least partly through ITCH activation.

Similarly, depletion of ITCH in melanocytes (Supplementary Fig. 4c) and BRAF^WT-expressing WM3918 (Fig. 3c), WM1346 (Fig. 3d, e), M245 (Supplementary Fig. 4d), and IPC-298 (Supplementary Fig. 4e) melanoma cells led to a remarkable

reduction of p-MEK and p-ERK levels. In line with the nondegradable function of K27-linked polyubiquitination, ITCH knockdown did not affect BRAF abundance (Fig. 3a–e), while it abolished the polyubiquitination of endogenous BRAF (Fig. 3f; Supplementary Fig. 4f, g). On the other hand, although BRAF ubiquitination is attenuated in *ITCH*-depleted, BRAF^V600E-expressing 1205Lu melanoma cells (Supplementary Fig. 4g), p-MEK and p-ERK levels in 1205Lu cells were not affected (Supplementary Fig. 4h). These results suggest that ITCH-mediated BRAF ubiquitination controls the activation of WT-BRAF, but not the hyperactive V600E-BRAF. Furthermore, reintroduction of WT-ITCH but not the catalytic inactive CS-ITCH, rescued the downregulated p-MEK/p-ERK in MEFs and

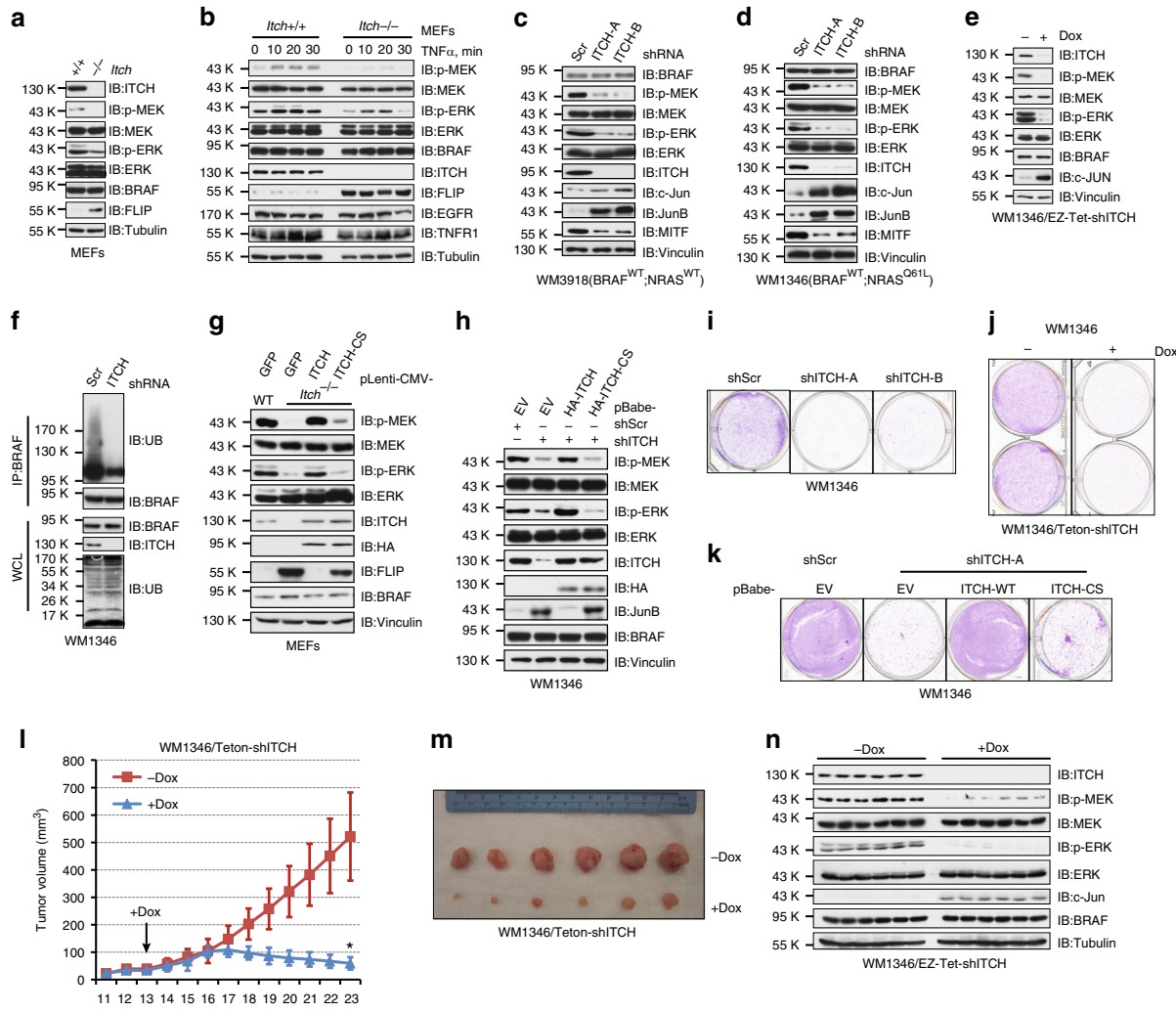

**Fig. 3** Depletion of ITCH attenuates BRAF activity and suppresses melanoma cell growth. **a** Immunoblot (IB) analysis of whole-cell lysates (WCL) derived from WT and *Itch*^-/- MEFs. **b** IB analysis of WCL derived from WT and *Itch*^-/- MEFs treated with 50 ng•ml^-1 TNFα for the indicated time period. **c, d** IB analysis of WCL derived from WM3918 (**c**) and WM1346 (**d**) cells infected with the indicated sh*ITCH* lentiviral constructs; a scrambled shRNA construct (shScr) was used as the negative control. **e** IB analysis of WCL derived from WM1346 cells stably expressing EZ-Tet-pLKO-sh*ITCH* which allows doxycycline-induced depletion of ITCH. Cells were treated with 1 μg•ml^-1 doxycycline (Dox), as indicated for 48 h before harvest. **f** Depletion of ITCH abolished endogenous BRAF ubiquitination. IB analysis of WCL and anti-BRAF immunoprecipitates (IP) derived from WM1346 cells infected with shScr or sh*ITCH* lentiviral construct. **g, h** WT- but not enzymatic dead C832S (CS)-ITCH rescued decreased p-MEK and p-ERK upon ITCH depletion. IB analysis of WCL derived from MEFs (**g**) and WM1346 (**h**) cells infected with the indicated viral constructs. Murine ITCH cDNA was used in the reconstitution experiments. **i–k** WM1346 cells generated in (**d**, **e**, **h**) were subjected to clonogenic survival assays in RPMI-1640 media supplemented with 10% FBS for 14 days. Crystal violet was used to stain the formed colonies (**i-k**), and representative pictures were shown from three independent experiments. **l, m** Tumor growth curves (**l**) and pictures at the end point (**m**) for the xenograft experiments with the WM1346 cells generated in (**e**) were inoculated subcutaneously. In each flank of six nude mice, 1 × 10^6 cells were injected. The visible tumors were measured at the indicated days. The arrow indicates the day doxycycline administration was started. Error bars represent ± SEM (n = 6). IB analysis of WCL derived from the tumor samples generated in (**l, m**)

WM1346 cells (Fig. 3g, h), indicating an E3-ligase activity-dependent manner of regulating MEK/ERK signaling by ITCH.

The BRAF/MEK/ERK pathway is crucial to melanoma cell growth (Supplementary Fig. 4i, j). In support of a potential role of ITCH in controlling BRAF function, we found that in melanoma cells, depletion of ITCH suppressed proliferation (Fig. 3i, j; Supplementary Fig. 4k–n) and 3D spheroid formation of WM1346 cells (Supplementary Fig. 4o, p). In addition, reconstituting shITCH-WM1346 cells with WT-ITCH, but not the catalytic inactive CS-ITCH, rescued the reduced cell growth (Fig. 3k; Supplementary Fig. 4q). Furthermore, doxycycline-induced ITCH depletion in WM1346 cells significantly suppressed the p-MEK/p-ERK signaling and tumor growth in vivo (Fig. 3l–n). Our results coherently demonstrate that ITCH positively regulates the BRAF/MEK/ERK signaling cascade and facilitates the survival of BRAF$^{WT}$ melanoma cells.

**Cytokine promotes BRAF ubiquitination and activation.** Proinflammatory molecules, such as TNFα, interleukin 1β (IL-1β), and lipopolysaccharide (LPS), activate ITCH through JNK-mediated phosphorylation at the linker region between C2 and WW domains[32]. These phosphorylation events lead to a conformational change that disrupts the inhibitory intramolecular interaction between the WW and the HECT domains and augments the catalytic activity of ITCH[32,33]. These studies prompted us to investigate if TNFα or IL-1β facilitates ITCH-dependent BRAF ubiquitination and subsequent activation. As shown in Fig. 4a, b and Supplementary Fig. 6a–d, treatment of BRAF$^{WT}$ melanoma cells with TNFα or IL-1β led to a rapid induction of p-MEK/p-ERK levels, which was accompanied with the activation of JNK, supporting the correlation between BRAF function and the JNK/ITCH pathway activation. Notably, ubiquitination of endogenous BRAF was induced by TNFα or IL-1β treatment in both melanocytes and melanoma cells (Fig. 4c; Supplementary Fig. 6e–h), while depletion of ITCH abrogated TNFα-stimulated BRAF ubiquitination (Supplementary Fig. 6i, j).

In accordance with the role of growth factors in facilitating ITCH-dependent MEK–ERK activation (Supplementary Figs. 4a, b, 6k), we found that EGF also promoted BRAF ubiquitination, albeit weaker, compared with TNFα (Supplementary Fig. 6l). Importantly, ITCH is indispensable for EGF-triggered BRAF ubiquitination (Supplementary Fig. 6m). It is noteworthy that although TNFα promoted BRAF ubiquitination in BRAF$^{V600E}$-expressing 1205Lu melanoma cells (Supplementary Fig. 6n), the treatment only moderately affected MEK/ERK activities (Supplementary Fig. 6o), which is consistent with the results that p-MEK/p-ERK levels were insensitive to ITCH knockdown (Supplementary Fig. 4h). To assess the role of ITCH in TNFα-triggered BRAF activation, we found that p-MEK and p-ERK were refractory to TNFα treatment in ITCH-depleted melanoma cells (Fig. 4d; Supplementary Fig. 6p). Further analysis of endogenous BRAF ubiquitination revealed that TNFα-mediated BRAF ubiquitination is abolished upon Itch deletion in MEFs (Fig. 4e).

JNK phosphorylates ITCH at S199, T222, and S232 to unleash the enzymatic activity of ITCH in response to upstream signals[34]. In line with this finding, we observed increased ITCH phosphorylation in melanoma cells after TNFα treatment (Supplementary Fig. 7a). Moreover, in the TCGA cutaneous melanoma (TCGA-SKCM) RPPA (reverse-phase protein array) dataset containing 354 tumors, we found a positive correlation ($r = 0.28$, $p = 5.4 \times 10^{-8}$) between pT183/pY185-JNK1/2 and pT202/pY204-ERK1/2 signals from this cohort of samples (Supplementary Fig. 7b).

Next, we sought to examine the role of JNK in ITCH-mediated BRAF activation. Similar to our observation in Itch$^{-/-}$ MEFs

(Fig. 3a), deletion of both Jnk1 and Jnk2 in MEFs led to reduced levels of p-MEK and p-ERK in parallel with inactivation of ITCH, which is supported by the accumulation of its substrate c-FLIP (Fig. 4f). Furthermore, knockdown of JNK1 and JNK2 in melanoma cells suppressed MEK and ERK activities, while BRAF protein levels remained unchanged (Fig. 4g). Notably, compared with WT-ITCH-expressing cells, melanoma cells harboring a JNK phosphorylation-deficient 3A (S199A/T222A/S232A)-ITCH exhibited a significant reduction of both BRAF and c-Jun ubiquitination upon TNFα stimulation, and were unresponsive to TNFα-triggered MEK/ERK activation (Supplementary Fig. 7c, d).

Proinflammatory cytokines have been demonstrated to facilitate melanoma cell proliferation and metastasis, as well as to contribute to BRAF inhibitor resistance[6]. We found that compared with control cells, melanoma cells treated with TNFα or IL-1β displayed increased cell growth (Fig. 4h; Supplementary Fig. 7e–j). One of the major sources of proinflammatory cytokines in the tumor environment is the tumor-associated macrophages, particularly the M2 macrophage[35]. By utilizing a recently developed melanoma cell and M2 macrophage coculture model[6] (Fig. 4i), we revealed that the M2-differentiated THP1 cells could promote the proliferation of WM3918 cells (Fig. 4j, k), whereas TNFα blocking antibody could largely suppress THP1 coculture-induced WM3918 proliferation (Supplementary Fig. 7k, l). Furthermore, compared with control cells, elevations of p-JNK, p-MEK, and p-ERK levels were observed in cells cocultured with M2 THP1 cells (Fig. 4l).

Proliferation of melanocytes in vitro requires growth factors or mitogens such as TPA (12-O-tetradecanoylphorbol-13-acetate), TPA-independent growth has been widely used as evidence for melanocyte transformation[18,36]. In accordance with a pro-survival role of ITCH in melanoma cells, we found that overexpression of ITCH activated MEK/ERK, and supported TPA-independent growth of melan-a cells (Fig. 4m–o). Moreover, although expression of 3D-ITCH alone failed to promote anchorage-independent growth of melan-a cells (Fig. 4p, q), when Pten was further depleted (Supplementary Fig. 7m–o), 3D-ITCH-expressing melan-a cells formed colonies in the soft agar (Fig. 4p, q). Together, our findings support the model that cytokine stimulation activates JNK and in turn facilitates ITCH activation to promote BRAF ubiquitination and subsequent elevation of MEK/ERK signals (Supplementary Fig. 7p).

**Ubiquitination-deficient BRAF fails to activate MEK/ERK.** To define if BRAF ubiquitination is a determining factor in controlling its downstream signaling, we compared p-MEK and p-ERK levels in HEK293 cells transfected with WT or numerous BRAF mutants with the lysine residues mutated to arginine (KR mutant), which were resistant to ITCH-mediated ubiquitination (Fig. 1k, l). Intriguingly, compared with WT-BRAF, all the KR mutants, except for K699R, displayed reduced activity to stimulate MEK/ERK activities (Fig. 5a). Consistent with the results shown in Supplementary Fig. 4h, we found that introducing 5KR mutations to BRAF$^{V600E}$ failed to suppress its activity in HEK293 cells (Supplementary Fig. 8a).

Impaired BRAF function with the ubiquitination-deficient 5KR mutations indicates that either the intrinsic BRAF kinase activity is affected due to conformational change or the regulatory mechanisms that control BRAF function interfere with the KR substitutions. To gain insight into the mechanism, our in vitro kinase assay results demonstrated that the KR mutants of BRAF demonstrated similar activity to catalyze GST-MEK1 phosphorylation (Fig. 5b; Supplementary Fig. 8b). Furthermore, BRAF$^{WT}$ and BRAF[5] exhibited identical kinase kinetics in promoting GST-MEK1

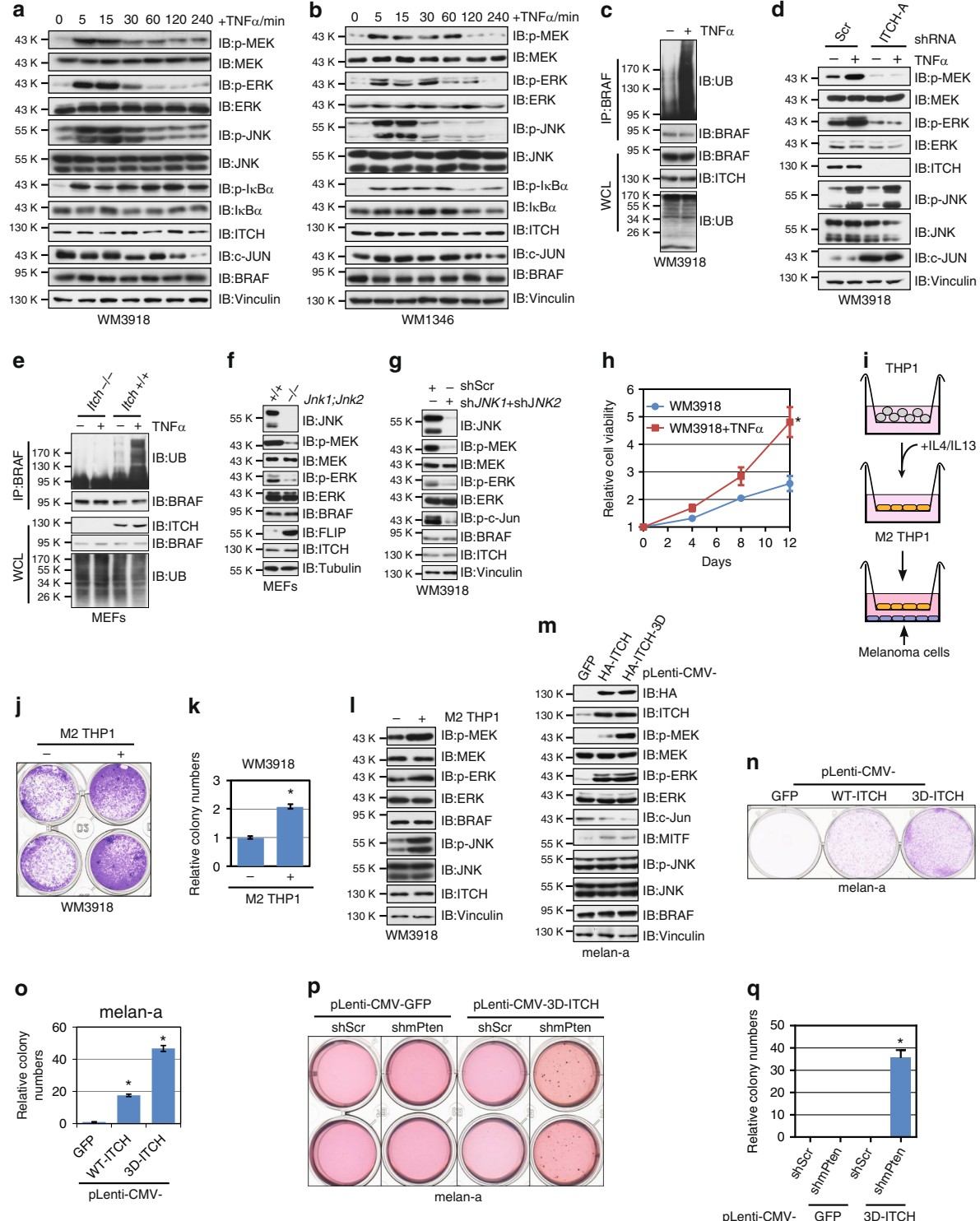

phosphorylation (Fig. 5c), which could be supported by similar circular dichroism spectra of the kinase domains from WT- and 5KR-BRAF (Supplementary Fig. 8c, d), suggesting that KR mutations barely affect BRAF protein conformation. More importantly, polyubiquitinated BRAF proteins displayed similar kinase kinetics compared with its unmodified counterpart (Fig. 5d).

Consistent with the results using ubiquitination-deficient BRAF mutant, we found that ITCH-binding-deficient BRAF S675A/P676A and P751A/P754A mutants also displayed reduced activity to propel MEK/ERK signaling in cells (Fig. 5e), while they were still capable of catalyzing GST-MEK1

phosphorylation in vitro (Fig. 5f). Next, we sought to determine if cells expressing ubiquitination-deficient 5KR-BRAF were refractory to cytokine stimuli. We generated HEK293 cell lines stably expressing WT-BRAF or 5KR-BRAF close to the endogenous level. As shown in Fig. 5g, compared with WT-BRAF-expressing cells, TNFα stimulation only minimally triggered p-MEK and p-ERK activation in 5KR-BRAF-expressing cells.

**Ubiquitination abolishes inhibitory BRAF/14–3–3 interaction.**
Given the non-proteolytic role of the K27-linkage ubiquitin chain,

**Fig. 4** Proinflammatory cytokines engage ITCH to promote BRAF ubiquitination and subsequent activation. **a**, **b** Immunoblot (IB) analysis of whole-cell lysates (WCL) derived from WM3918 (**a**) and WM1346 (**b**) cells treated with 50 ng•ml$^{-1}$ TNFα for the indicated time period. **c** Endogenous BRAF ubiquitination was increased after TNFα treatment. IB analysis of WCL and anti-BRAF immunoprecipitates (IP) derived from WM3918 cells treated with TNFα for 20 min. **d** IB analysis of WCL derived from shScr- and shITCH-WM3918 cells. **e** Endogenous BRAF ubiquitination was elevated in WT, but not Itch$^{-/-}$. **f** IB analysis of WCL derived from WT and Jnk1$^{-/-}$;Jnk2$^{-/-}$ MEFs. **g** IB analysis of WCL derived from WM3918 cells co-infected with shJNK1 and shJNK2 lentiviral constructs. **h** WM3918 cells were subjected to cell proliferation assays for 12 days. Cell viability was determined at the indicated time points. The viability was calculated as mean ± SD ($n = 3$) from three independent experiments. *$P < 0.05$; Student's $t$ test. **i** Illustration of the coculture experiment of M2-differentiated THP1 cells and melanoma cells. **j**, **k** Coculture with M2- differentiated THP1 cell stimulated WM3918 cells growth. The colony numbers (**j**) were calculated as mean ± SD ($n = 3$), *$P < 0.05$; Student's $t$ test (**k**). **l** Coculture with M2-differentiated THP1 cell activated the MEK/ERK signaling in WM3918 cells. IB analysis of WCL derived from WM3918 cells of the coculture experiment as described in (**j**, **k**). **m** IB analysis of WCL derived from melan-a cells stably expressing GFP, WT-ITCH of the constitutively active 3D-ITCH. **n**, **o** Melan-a cells generated in (**m**) were subjected to clonogenic survival assays without TPA for 14 days. The colony numbers were calculated as mean ± SD ($n = 3$), *$P < 0.05$; Student's $t$ test (**o**). **p**, **q** Melan-a cells generated in (**m**) were transduced with shScr or shPten lentiviral constructs followed by soft agar colony-formation assays without TPA for 21 days (**p**). The colony numbers were calculated as mean ± SD ($n = 3$) from three independent experiments. *$P < 0.05$; Student's $t$ test (**q**)

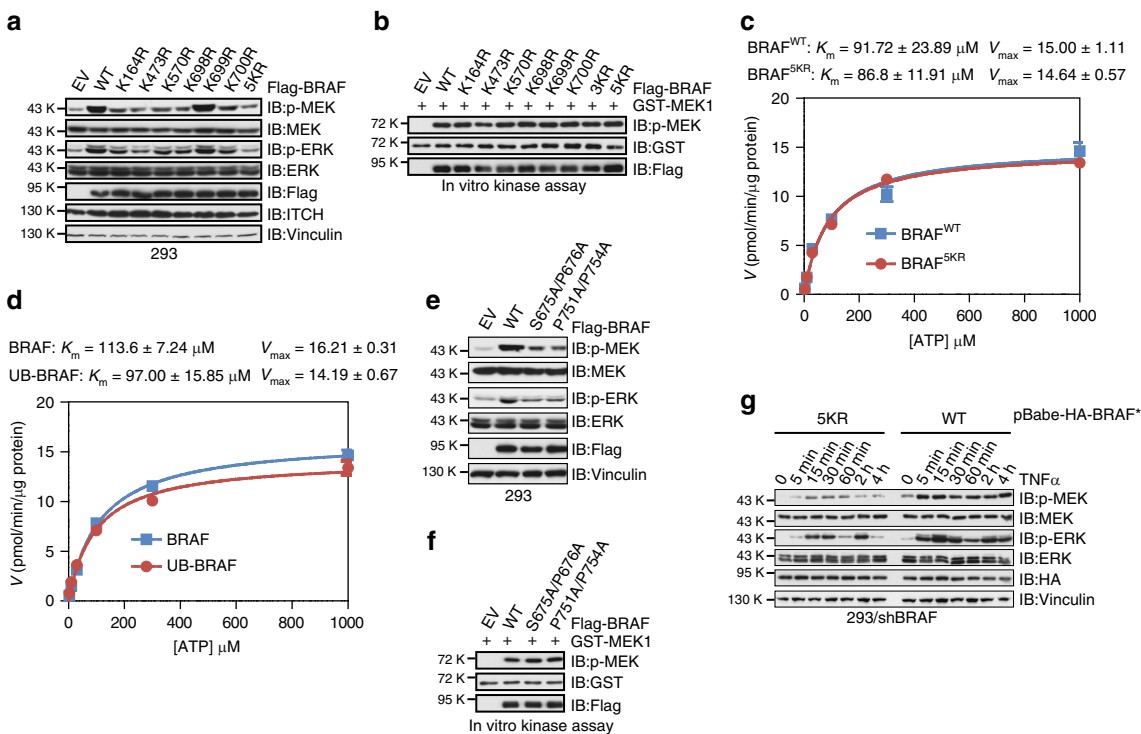

**Fig. 5** Ubiquitination-deficient BRAF mutants exhibit compromised activity. **a** Immunoblot (IB) analysis of whole-cell lysates (WCL) derived from HEK293 cells transfected with the indicated Flag-tagged WT-BRAF or BRAF-KR mutants. **b** In vitro kinase assays showing that immunopurified Flag-WT-BRAF and BRAF-KR proteins displayed similar activity to promote the phosphorylation of GST-MEK1. BRAF/3KR: K698R + K699R + K700R; BRAF/5KR: K164R + K473R + K570R + K698R + K700R. **c**, **d** Kinase kinetics of WT-BRAF and 5KR-BRAF (**c**) as well as unmodified and ubiquitinated BRAF proteins (**d**). Initial rates were measured at various concentrations of ATP using the continuous assay. The rates were replotted against substrate concentration and fit to the Michaelis–Menten equation. **e** IB analysis of WCL derived from HEK293 cells transfected with the indicated Flag-tagged WT-BRAF or ITCH-binding-deficient BRAF mutants. **f** In vitro kinase assays showing that immunopurified Flag-WT-BRAF and ITCH-binding-deficient BRAF proteins displayed similar activity to promote the phosphorylation of GST-MEK1. **g** IB analysis of WCL derived from HEK293 cells stably expressing WT-BRAF or 5KR-BRAF treated with 50 ng•ml$^{-1}$ TNFα for the indicated time period

we hypothesized that the K27-linked polyubiquitin chains assembled on BRAF serve a scaffolding role, a function that has been well characterized for K63-linked and linear polyubiquitin chains[15]. We compared the binding of unmodified and ubiquitinated BRAF with a panel of its upstream regulators and downstream effectors. As shown in Fig. 6a, the interaction of ubiquitinated BRAF with 14–3–3 was reduced compared with unmodified BRAF. In contrast, bindings of BRAF with MEK1 (Supplementary Fig. 9a), NRAS (Supplementary Fig. 9b), BRAF (Supplementary Fig. 9c), CRAF (Supplementary Fig. 9d), and KSR1 (Supplementary Fig. 9e), were not affected. Furthermore,

we found that while the interaction between 14–3–3 and WT-BRAF was attenuated after ubiquitination, the binding between 14–3–3 and 5KR-BRAF or ITCH-binding-deficient BRAF was not affected (Fig. 6b; Supplementary Fig. 9f). Moreover, the binding of endogenous BRAF with 14–3–3 proteins was attenuated after melanoma cells were treated with TNFα (Fig. 6c; Supplementary Fig. 9g), which were shown to activate JNK/ITCH to facilitate BRAF ubiquitination (Fig. 4a–c; Supplementary Fig. 6f).

Previous studies demonstrated a pivotal role for 14–3–3 proteins in controlling the activation of RAF proteins, including

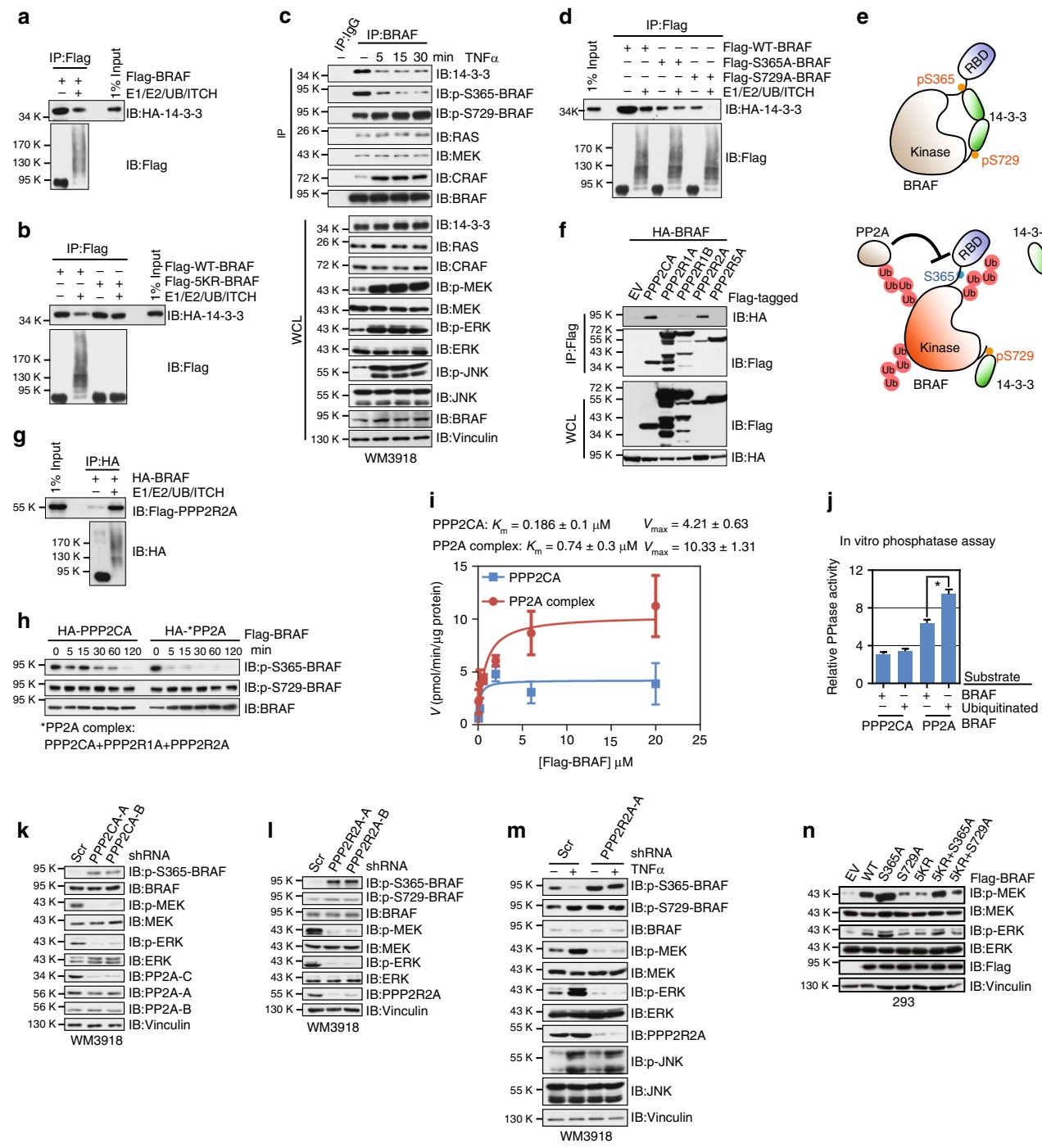

BRAF. In total, 14–3–3 proteins recognize RSxpSxP or RxxxpSxP motifs, where pS represents a phospho-serine[37]. p-S365 and p-S729 are 14–3–3 interacting sites found in BRAF[38]. Notably, 14–3–3 proteins play opposing roles in regulating BRAF protein activity (Supplementary Fig. 9h). The intramolecular interaction of p-S365 and p-S729 sites mediated by the 14–3–3 dimer enforces a closed, inactive conformation of BRAF, while the intermolecular dimerization of two BRAF proteins is stabilized by the binding of the 14–3–3 dimer via the p-S729 sites at both BRAF proteins[39].

To assess the impact of BRAF ubiquitination on its binding to the 14–3–3 proteins at different sites[39], we found that mutation of either S365 or S729 resulted in a reduced binding with 14–3–3 (Fig. 6d). Intriguingly, although both S365A-BRAF and S729A-BRAF were ubiquitinated in a similar fashion as their WT counterpart, only ubiquitinated WT and S729A-BRAF, but not ubiquitinated S365A-BRAF, displayed reduced binding with 14–3–3 (Fig. 6d). This result hence underpins the model that the K27-linked polyubiquitination specifically interferes with the binding between 14–3–3 and p-S365, but not p-S729 (Fig. 6e).

Since the binding of 14–3–3 to the C-terminus p-S729 sites promotes RAF protein dimerization[40], we next sought to determine if BRAF polyubiquitination controls BRAF dimerization. As shown in Supplementary Fig. 9c and d, the binding between RAF proteins was not affected in vitro. Moreover, the dimerization-compromised S729A-BRAF mutant also exhibited a reduced binding with 14–3–3, due to lack of the S729 phosphorylation (Supplementary Fig. 9i). More importantly,

**Fig. 6** Ubiquitination of BRAF disrupts the inhibitory interaction between BRAF and 14–3–3 proteins. **a** Ubiquitinated BRAF displayed reduced binding with 14–3–3 in vitro. In vitro ubiquitination assays using immunopurified Flag-BRAF proteins were performed followed by incubation of Flag-BRAF with 14–3–3. The anti-Flag immunoprecipitates (IP) were subjected to SDS-PAGE and immunoblot (IB) analysis. **b** 5KR-BRAF displayed stronger binding to 14–3–3 compared with WT-BRAF. **c** Binding between endogenous BRAF and 14–3–3 was reduced upon TNFα treatment. **d** S365A-BRAF and S729A-BRAF mutants exhibited a distinct affinity with 14–3–3 after ubiquitination. **e** Illustration of the proposed models for the disruption of BRAF-14–3–3 interaction by K27-linked ubiquitination. **f** PPP2CA and PPP2R2A subunits of the PP2A protein family specifically interacted with BRAF. **g** Ubiquitinated BRAF displayed increased binding with PPP2R2A in vitro. **h** In vitro phosphatase assay showing that compared with the catalytic PPP2CA subunit, the PP2A complex was more active in dephosphorylating p-S365-BRAF. Immunopurified Flag-BRAF proteins were incubated with the indicated PP2A phosphatases for the indicated time period before SDS-PAGE and IB analysis. **i** Phosphatase kinetics of PPP2CA and the PP2A complex in dephosphorylating BRAF. Initial rates were measured at various concentrations of Flag-BRAF protein using the continuous assay. The rates were replotted against substrate concentration and fit to the Michaelis–Menten equation. **j** In vitro phosphatase assay showing that ubiquitinated BRAF was a better substrate for the PP2A complex. Immunopurified Flag-BRAF were subjected to in vitro ubiquitination as described in Fig. 1c. Unmodified and ubiquitinated proteins were incubated with the indicated PP2A phosphatases for 60 min. The free phosphate was measured using the malachite green detection reagent, normalized, and calculated as mean ± SD (*n* = 3) from three independent experiments. *P < 0.05; Student's *t* test. **k, l** IB analysis of WCL derived from WM3918 cells infected with the indicated sh*PPP2CA* (**k**) or sh*PPP2R2A* (**l**) lentiviral constructs. **m** IB analysis of WCL derived from WM3918 cells infected with shScr or sh*PPP2R2A* lentiviral constructs. **n** IB analysis of WCL derived from HEK293 cells transfected with the indicated Flag-tagged WT-BRAF or BRAF mutants

ubiquitinated S729A-BRAF almost completely lost its binding to 14–3–3 (Supplementary Fig. 9i), further supporting a role of BRAF atypical ubiquitination in specifically regulating BRAF intramolecular interaction, but not BRAF–BRAF or BRAF–CRAF dimerization.

Although polyubiquitin chains conjugated to BRAF could block the binding between 14–3–3 and BRAF in vitro (Fig. 6a), given the low stoichiometry of polyubiquitinated BRAF observed in cells (Fig. 4c, e; Supplementary Fig. 6e–j), it is inconceivable that the displacement of 14–3–3 from ubiquitinated BRAF is the primary mechanism that accounts for the decreased 14–3–3/BRAF binding in cells (Fig. 6c; Supplementary Fig. 9g). Intriguingly, accompanied by reduced 14–3–3 binding, BRAF p-S365 phosphorylation was significantly downregulated in melanoma cells treated with TNFα (Fig. 6c; Supplementary Fig. 9g), suggesting that K27-linked ubiquitination of BRAF may facilitate the dephosphorylation of p-S365.

PP2A family phosphatases have been shown to dephosphorylate pS259-CRAF[41–43]. The PP2A holoenzyme consists of three subunits, a scaffolding subunit A, a catalytic subunit C, and one of the four regulatory subunits B (B55/PR55), B' (B56/PR61), B" (PR48/PR72/PR130), or B"'(PR93/PR110)[44] (Supplementary Fig. 9j[45]). Notably, the catalytic PPP2CA subunit and the B55 subfamily regulatory subunit PPP2R2A exhibited a strong binding with BRAF, among a number of PP2A family components examined (Fig. 6f). Different from the B56 subfamily, the B55 subfamily regulatory subunits are WD40-repeat domain-containing proteins (Supplementary Fig. 9j). WD40-repeat domain mediates protein–protein interactions, and WD40-repeat domain-containing proteins have been shown to bind ubiquitin[46]. Moreover, a recent proteomic study revealed PPP2R2A as a BRAF-interacting protein in cells[47]. Thus, we postulate that the K27-linked polyubiquitin chain might recruit the PP2A phosphatase to dephosphorylate p-S365, leading to further reduced binding with 14–3–3 in cells (Fig. 6e). In support of this notion, compared with unmodified BRAF, ubiquitinated BRAF displayed a stronger binding to the PPP2R2A subunit (Fig. 6g), but not the PPP2CA catalytic subunit (Supplementary Fig. 9k).

In vitro phosphatase assays demonstrate that compared with the catalytic subunit PPP2CA, the PP2A complex encompassing PPP2CA, PPP2R2A, and PPP2R1A displayed enhanced activity in dephosphorylating BRAF (Fig. 6h, i; Supplementary Fig. 9l). Importantly, the PP2A complex accelerated the dephosphorylation of p-S365-, but not p-S729-BRAF (Fig. 6h). Moreover, ubiquitinated BRAF appeared to be a better substrate for the PP2A complex containing the ubiquitin-binding PPP2R2A subunit (Fig. 6j). In support of a key role for PP2A in regulating

BRAF activity, we found that in melanoma cells, depletion of PPP2CA or PPP2R2A led to increased p-S365-BRAF and decreased MEK/ERK activity (Fig. 6k, l; Supplementary Fig. 9m). More importantly, TNFα failed to activate BRAF in PPP2R2A-depleted melanoma cells (Fig. 6m; Supplementary Fig. 9n).

In line with previous reports, we found that compared with WT-BRAF, S365A-BRAF exhibited a higher activity while S729A-BRAF was less active (Fig. 6n). Notably, introducing the S365A mutation to the non-ubiquitinatable 5KR-BRAF or ITCH-binding-deficient BRAF largely restored its activity in activating MEK (Fig. 6n; Supplementary Fig. 9o). Consistently, compared with WT-BRAF, non-ubiquitinatable 5KR-BRAF and ITCH-binding-deficient BRAF exhibited elevated S365 phosphorylation in cells (Supplementary Fig. 9p). Together, our results illustrate a unique role of BRAF K27-linked ubiquitination in disrupting 14–3–3-mediated BRAF inhibition.

**Ubiquitination-deficient BRAF is less tumorigenic**. We next sought to functionally characterize if the ubiquitination-deficient BRAF mutant exhibits a compromised function in maintaining the growth and invasion of BRAF[WT]-expressing melanoma cells. To this end, we found that compared with WT-BRAF, ubiquitination-deficient 5KR-BRAF failed to promote MEK/ERK activation in *BRAF*-depleted melanoma cells (Fig. 7a; Supplementary Fig. 10a–b). Furthermore, compared with WT-BRAF, 5KR-BRAF-expressing WM1346 cells were refractory to TNFα-mediated MEK/ERK activation (Supplementary Fig. 10c), which was accompanied with decreased ubiquitination (Fig. 7b; Supplementary Fig. 10d, e).

Reconstitution of BRAF[WT] in *BRAF*-depleted WM3918 and WM1346 cells promoted cell proliferation (Fig. 7c, d; Supplementary Fig. 11a–f). In contrast, BRAF[5KR] exhibited reduced activity to promote cell proliferation in both cell lines (Fig. 7c, d; Supplementary Fig. 11a–f), suggesting a crucial role for K27-linked polyubiquitination to maintain BRAF activity in melanoma cells. Moreover, BRAF[5KR]-expressing WM1346 cells exhibited a reduced invasion into the Matrigel (Fig. 7e, f) and a layer of endothelial cells (Fig. 7g, h). In line with our observation that M2-differentiated macrophages promoted MEK/ERK activation and subsequent proliferation of WM3918 cells (Fig. 4i–k), M2-differentiated THP1 cells stimulated cell growth in BRAF[WT]-expressing WM3918 cells, while the effect was compromised in BRAF[5KR]-expressing cells (Fig. 7i, j).

The attenuated cell proliferation and invasion in BRAF[5KR]-expressing cells were further supported by the anchorage-independent soft agar colony-formation experiments. Compared

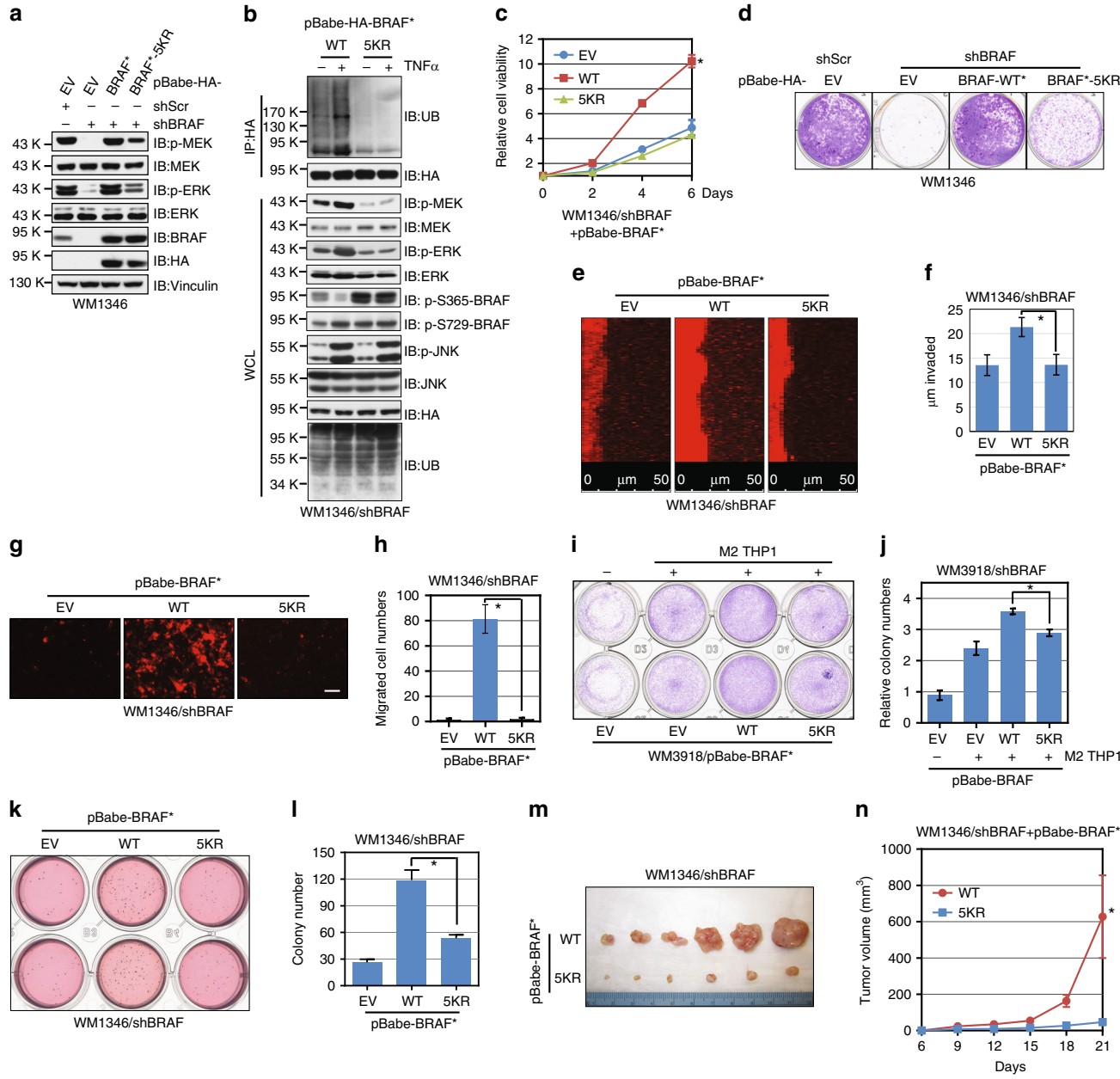

**Fig. 7** Ubiquitination of BRAF serves to maintain BRAF oncogenic function in melanoma cells. **a** Immunoblot (IB) analysis of whole-cell lysates (WCL) derived from WM1346 cells stably expressing EV, WT-BRAF, or the 5KR-BRAF mutant. The stable cell lines were further infected with the indicated lentiviral vectors to deplete endogenous *BRAF*. *The BRAF cDNA used in this figure was mutated to confer resistance to the sh*BRAF* lentiviral vector. **b** IB analysis of anti-BRAF immunoprecipitates (IP) BRAF-depleted WM1346 cells stably expressing WT-BRAF or the 5KR-BRAF mutant. **c** WM1346 cells generated in (**a**) were subjected to cell proliferation assays. Cell viability was calculated as mean ± SD ($n = 3$) from three independent experiments. *$P < 0.05$; Student's *t* test. **d** WM1346 cells generated in (**a**) were seeded (300 cells per well) for clonogenic survival assays. **e, f** WM1346 cells generated in (**a**) were subjected to Matrigel invasion assays (**e**). The invaded distance was calculated as mean ± SD ($n = 3$) from three independent experiments. *$P < 0.05$; Student's *t* test (**f**). **g, h** WM1346 cells generated in (**a**) were subjected to endothelial cell invasion assays (**g**). The invaded distance was calculated as mean ± SD ($n = 3$) from three independent experiments. *$P < 0.05$; Student's *t* test (**h**). Scale bar: 50 μm. **i, j** WM3918 cells generated in Supplementary Fig. 11a were subjected to clonogenic survival assays without or with M2-differentiated THP1 (**i**). The colony numbers were counted from three independent experiments. The colony numbers were calculated as mean ± SD ($n = 3$), *$P < 0.05$; Student's *t* test (**j**). **k, l** WM1346 cells generated in (**a**) were subjected to soft agar colony-formation assays (**k**). The colony numbers were counted and calculated as mean ± SD ($n = 3$) from three independent experiments. *$P < 0.05$; Student's *t* test (**l**). **m, n** Tumor pictures (**m**) and the growth curves (**n**) for the xenograft experiments with the WM1346 cells generated in (**a**) were inoculated subcutaneously. The visible tumors were measured at the indicated days. Error bars represent ± SEM ($n = 6$). *$P < 0.05$; Student's *t* test

with BRAF$^{WT}$, expression of BRAF$^{5KR}$ only minimally promoted colony growth in soft agar (Fig. 7k, l; Supplementary Fig. 11g, h). Furthermore, while the expression of BRAF$^{WT}$ promoted tumor growth in immunodeficient mice, WM1346 cells bearing BRAF$^{5KR}$ were less tumorigenic in vivo (Fig. 7m, n; Supplementary Fig. 11i, j). The slower tumor growth in BRAF$^{5KR}$-expressing cells was supported by diminished MEK/ERK signaling in tumor tissue samples (Supplementary Fig. 11k, l). Our results collectively demonstrate that compared with BRAF$^{WT}$, which facilitates melanoma cell proliferation and invasion, the ubiquitination-deficient BRAF$^{5KR}$ mutant is less active and thus exhibited compromised oncogenic functions. These data thus support a post-translational modification by atypical ubiquitination of BRAF to sustain the MEK/ERK signaling cascade in melanoma cells.

## Discussion

Aberrant activation of the RAS/RAF signaling pathway is a hallmark in human cancers[38]. Recent advances highlighted the crosstalk between the TME and the MAPK signaling[48]. Our findings here present a molecular switch that in response to cytokine stimuli, the JNK kinase activates the ITCH ubiquitin E3 ligase, which in turn catalyzes the K27-linked polyubiquitination of BRAF. K27-linkage polyubiquitin chains formed on BRAF serve as a barrier to preclude 14–3–3 from binding to BRAF at the p-S365 site, and as a scaffold to recruit the PP2A phosphatase to dephosphorylate p-S365 in cells, thereby sustaining BRAF activity to drive downstream MEK/ERK signals (Supplementary Fig. 11m).

Our results suggest that ubiquitination-deficient 5KR-BRAF maintains the same conformation as its WT counterpart (Supplementary Fig. 8d), and thus displays a comparable activity in phosphorylating MEK1 in vitro (Fig. 5b). On the other hand, 5KR-BRAF appeared to be less active in cells (Figs. 5a, g, 7a; Supplementary Fig. 10a–c). The decreased kinase activity for 5KR-BRAF in cells was likely due to the increased inhibitory S365 phosphorylation (Fig. 7b; Supplementary Fig. 10d). However, immunopurified BRAF proteins used in our in vitro kinase assays (Fig. 5b) exhibited similar levels of S365 phosphorylation (Supplementary Fig. 8e). We reasoned that immunopurified BRAF proteins from 293T cells were highly overexpressed, leading to minimal post-translational modifications. In support of this notion, we found that although compared with 5KR-BRAF, WT-BRAF displayed a higher activity when expressed at a level comparable with the endogenous one, both BRAF species exhibited similar activity when highly overexpressed in cells (Supplementary Fig. 8f). Furthermore, compared with HA-5KR-BRAF, immunopurified HA-WT-BRAF proteins from stable cell lines (Fig. 7a) were more active in phosphorylating MEK1 in vitro (Supplementary Fig. 8g).

We observed that compared with BRAF$^{WT}$, BRAF$^{V600E}$ exhibited higher basal ubiquitination and appeared to be a better substrate for ITCH (Supplementary Fig. 1e). The V600E substitution in BRAF favors an open and active conformation[49], which might expose more lysines for ubiquitination. It was found that ITCH deficiency abolished the ubiquitination of both BRAF$^{WT}$ and BRAF$^{V600E}$ (Supplementary Fig. 1h), supporting a role for endogenous ITCH in maintaining a higher level of BRAF$^{V600E}$ basal ubiquitination. ITCH has been shown to promote its substrate polyubiquitination through different linkages[26–29]. In further support of the notion that ITCH promotes c-Jun and BRAF ubiquitination through K48- and K27-linkages, respectively (Supplementary Fig. 1g), we found that BRAF appeared to be very stable, regardless of *Itch* genetic status (Supplementary Fig. 5a) or ITCH activity (Supplementary Fig. 5b,

c). On the contrary, c-Jun exhibited an extended half-life in melanoma cells harboring the inactive CS-ITCH mutant (Supplementary Fig. 5b), while becoming less stable in cells stimulated with TNFα (Supplementary Fig. 5c).

ITCH has been initially identified as a critical player in immune response and was therefore named after the itchy phenotype observed in *Itch*-deficient mice[50]. Further investigations revealed that ITCH controls T-lymphocyte differentiation by targeting Jun family transcription factors for degradation[51,52]. We demonstrate that ITCH facilitates the survival of BRAF$^{WT}$-expressing melanoma cells (Fig. 3i–m; Supplementary Fig. 4k–n), supporting an oncogenic function of ITCH in BRAF$^{WT}$ melanoma. It is worthy to note that although ITCH knockdown suppressed p-MEK and p-ERK in melanocytes (Supplementary Fig. 4c), similar to BRAF knockdown (Supplementary Fig. 5d), it only moderately inhibited melanocyte growth (Supplementary Fig. 5e–g). The addiction of melanoma cells to the ITCH–BRAF signaling indicates that ITCH could be a potent therapeutic target for BRAF$^{WT}$-expressing melanomas.

## Methods

**Plasmids**. pCINeo-myc-Itch (#11427), pRK5-HA-Ubiquitin-WT (#17608), pRK5-HA-Ubiquitin-K0 (#17603), pRK5-HA-Ubiquitin-K6 (#22900), pRK5-HA-Ubiquitin-K11 (#22901), pRK5-HA-Ubiquitin-K27 (#22902), pRK5-HA-Ubiquitin-K29 (#22903), pRK5-HA-Ubiquitin-K33 (#17607), pRK5-HA-Ubiquitin-K48 (#17605), pRK5-HA-Ubiquitin-K63 (#17606), pBABE zeo PPP2CA (#10689), pMIG-PP2A-Abeta (#15248), pMIG FLAG B55 alpha (#13804), pMIG-Aalpha WT (#10884), V245 pCEP-4HA B56alpha (#14532), and EZ-Tet-pLKO-Hygro (#85972) were obtained from Addgene. pFlag-CMV2-BRAF, pFlag-CMV2-BRAF$^{V600E}$, pcDNA3-HA-CRAF, pCGN-HA-NRAS, and pcDNA3-HA-14–3–3–γ were obtained from Dr. Wenyi Wei[18]. BRAF and its mutant cDNAs were sub-cloned into pcDNA3-HA, pFlag-CMV2, pGEX-4T-1, pTRIPZ-CMV, and pBabe-hygro-HA vectors. ITCH and its C832S mutant were subcloned into pFlag-CMV2, pcDNA3-HA, pGEX-4T-1, and pLenti-CMV-hygro vectors. Ubiquitin and its mutants were subcloned into the pcDNA3–6His vector. pFlag-CMV2-ARAF and pFlag-CMV2-CRAF were described previously[18]. pFlag-CMV2-Smurf1, pFlag-CMV2-Smurf2, pFlag-CMV2-WWP1, pFlag-CMV2-WWP2, pFlag-CMV2-NEDD4, and pFlag-CMV2-NEDD4L were described previously[53]. shRNA constructs targeting human ITCH (RHS4533-EG83737) were purchased from Open-Biosystems. The targeting sequence of sh*JNK1* is 5′-gcagaagcaagcgtgacaaca-3′, the targeting sequence of sh*JNK2* is 5′-aaaggttgtgtgatattccaa-3′, the targeting sequence of sh*BRAF* is 5′-cagatgaagatcatcgaaat-3′, the targeting sequence of sh*PPP2CA-A* is 5′-tggaacttgacgatactctaa-3′, the targeting sequence of sh*PPP2CA-B* is 5′-accggaatgtagtaacgattt-3′, the targeting sequence of sh*PPP2R2A-A* is 5′-tatgtctcgctttgtgtttct-3′, and the targeting sequence of sh*PPP2R2A-B* is 5′-aaat-gacctgttactgggatc-3′. All the primers used in the study could be found in Supplementary Table 1.

**Antibodies**. Anti-BRAF (9433, 1:2000), anti-CRAF (9422, 1:2000), anti-pT202/pY204-ERK1/2 (4370, 1:3000), anti-ERK1/2 (4695, 1:3000), anti-pS217/pS221-MEK1/2 (9154, 1:3000), anti-MEK1/2 (9122, 1:3000), anti-SAPK/JNK (9252, 1:2000), anti-pT183/pY185-SAPK/JNK (4668, 1:2000), anti-c-Jun (9165, 1:2000), anti-JunB (3753, 1:2000), anti-ubiquitin (3933, 1:1000), anti-RAS (3339, 1:2000), anti-p-S259-CRAF (9421, 1:2000), anti-K48-linkage ubiquitin (D9D5, 1:2000), Human TNFα-neutralizing (D1B4) mAb (7321), and anti-GST (2622, 1:2000) antibodies were purchased from Cell Signaling Technology. Anti-FLIP (H-150, 1:2000), anti-pS63-c-Jun (KM-1, 1:2000), anti-p-S621-Raf-1 (E-1, 1:2000), anti-c-Myc (9E10, 1:2000), and polyclonal anti-HA (Y-11, 1:2000) antibodies were purchased from Santa Cruz. Anti-tubulin (T-5168, 1:2000) and anti-vinculin (V-4505, 1:2000) antibodies were purchased from Bethyl Labs. Polyclonal anti-Flag antibody (F-2425, 1:2000), monoclonal anti-Flag (F-3165, 1:2000) antibody, anti-Flag agarose beads (A-2220), and anti-HA agarose beads (A-2095), as well as peroxidase-conjugated anti-mouse secondary antibody (A-4416, 1:3000) and peroxidase-conjugated anti-rabbit secondary antibody (A-4914, 1:3000) were purchased from Sigma. Anti-K27-linkage ubiquitin (ab181537, 1:2000) was purchased from Abcam. Monoclonal anti-HA antibody (MMS-101P, 1:2000) was purchased from Covance.

**Cell culture, transfection, and infection**. In total, 293T, HEK293, and A375 cell lines were obtained from ATCC and grown in the Dulbecco's modified Eagle medium (DMEM) supplemented with 10% fetal bovine serum (FBS), 2 mM glutamine, 100 units·ml$^{-1}$ penicillin, and 100 mg·ml$^{-1}$ streptomycin. In total, 1205Lu, WM1346, and WM3918 cell lines were a gift from Dr. Meenhard Herlyn (The Wistar Institute), M245 was a gift from Dr. Antoni Ribas (UCLA Medical Center). IPC-298 was purchased from the Leibniz-Institut DSMZ (Germany). Melan-a cell

line was a gift from Dr. Wenyi Wei (BIDMC, Harvard Medical School). In total, 1205Lu, WM1346, WM3918, melan-a, M245, and IPC-298 cell lines were grown in the RPMI-1640 medium supplemented with 10% FBS, 2 mM glutamine, 100 units·ml$^{-1}$ penicillin, and 100 mg·ml$^{-1}$ streptomycin. For melan-a cell culture, 100 nM TPA (12-O-tetradecanoylphorbol-13-acetate) was added. Immortalized *Jnk1*, *Jnk2* double-knockout (*Jnk1*$^{-/-}$;*Jnk2*$^{-/-}$) MEFs and WT counterparts were a kind gift from Dr. Roger J. Davis. Immortalized *Itch* knockout (*Itch*$^{-/-}$) MEFs and WT counterparts were a kind gift from Drs. Lydia E. Matesic and Edward W. Harhaj. All MEFs were grown in DMEM supplemented with 10% FBS, 2 mM glutamine, 100 units·ml$^{-1}$ penicillin, and 100 mg·ml$^{-1}$ streptomycin. All cell lines were routinely tested to be negative for mycoplasma contamination.

**Site-directed mutagenesis.** Site-directed mutagenesis to generate BRAF, ITCH, and ubiquitin mutants was performed using the QuikChange XL Site-Directed Mutagenesis Kit (Agilent) according to the manufacturer's instructions.

**Lentiviral and retroviral packaging and infection.** Lentiviral constructs were co-transfected with the pCMV-dR8.91 (Delta 8.9) plasmid containing *gag*, *pol*, and *rev* genes, and the VSV-G envelope-expressing plasmid into 293T cells. For packaging retrovirus, retroviral constructs were co-transfected with VSV-G, JK3, and pCMV-tat into 293T cells. Virus-containing media were collected and filtered before being used for infection[54].

**Immunoblots and immunoprecipitation.** Cells were lysed in EBC buffer (50 mM Tris, pH 7.5, 120 mM NaCl, and 0.5% NP-40) supplemented with protease inhibitors (Thermo Scientific) and phosphatase inhibitors (Thermo Scientific). To prepare the whole-cell lysates (WCL), 3 × SDS sample buffer was directly added to the cell lysates and sonicated before being resolved on SDS-PAGE and subsequently immunoblotted with primary antibodies. The protein concentrations of the lysates were measured using the Bio-Rad protein assay reagent on a Bio-Rad Model 680 Microplate Reader. For immunoprecipitation, 1 mg of lysates were incubated with the appropriate agarose-conjugated primary antibody for 3–4 h at 4 °C or with an unconjugated antibody (1–2 µg) overnight at 4 °C, followed by 1-h incubation with Protein G Sepharose beads (GE Healthcare). Immunocomplexes were washed four times with NETN buffer (20 mM Tris, pH 8.0, 100 mM NaCl, 1 mM EDTA, and 0.5% NP-40) before being resolved by SDS-PAGE and immunoblotted with the indicated antibodies. Cells were pretreated with 10 µM MG132 for 12 h before harvest for IP analyses in Figs. 1a, b, e–g, k, 2a, b, e, i, and j. The uncropped scans of the blots were found in Supplementary Fig. 12.

**In vitro binding assays.** The pGEX-4T-1-Itch plasmid was transformed into BL21-(DE3) competent cells. The recombinant GST-ITCH proteins were expressed by isopropyl β-D-1-thiogalactopyranoside (IPTG) induction for 18 h at 16 °C. The proteins were purified using Glutathione Sepharose 4B (GE Healthcare) according to the manufacturer's instructions. Agarose-bound GST and GST-ITCH proteins were further incubated with in vitro transcribed and translated HA-BRAF using TnT Quick Coupled Transcription/Translation System from Promega. GST-pull-down experiments were also performed by incubating GST-fusion proteins with cell lysates from 293T cells expressing the indicated exogenous proteins. The incubation was performed at 4 °C for 3–4 h, followed by washing with NETN buffer, as described in the "Immunoblots and immunoprecipitation" section above. Samples were resolved by SDS-PAGE and subjected to immunoblot analysis.

**In vitro kinase assays.** BRAF was immunopurified from 293T cells transfected with Flag-BRAF constructs. In total, 0.5 M NaCl was used in the washing buffer to remove BRAF-associated proteins, the purity of Flag-BRAF proteins was confirmed by SDS-PAGE and Coomassie blue staining. GST-MEK1 was expressed in BL21 (DE3) *E. coli* and purified using Glutathione Sepharose 4B (GE Healthcare). BRAF kinase was incubated with 0.2 µg of GST-MEK1 in kinase assay buffer (10 mM HEPES, pH 8.0, 10 mM MgCl$_2$, 1 mM dithiothreitol, and 0.1 mM ATP). The reaction was initiated by the addition of GST-MEK1 in a volume of 30 µl for 15 min at 30 °C, followed by the addition of SDS-PAGE sample buffer to stop the reaction before being resolved by SDS-PAGE.

For kinase kinetics measurement, 0.5 µM purified BRAF protein was incubated with 1 µM GST-MEK1 and the indicated concentration of ADP-free ATP in kinase assay buffer (10 mM HEPES, pH 8.0, 10 mM MgCl$_2$, and 1 mM dithiothreitol) for 30 min at room temperature. The generated ADP was determined using ADP-Glo Kinase Assay Kit from Promega.

**In vitro phosphatase assays.** The PPP2CA catalytic subunit or the PP2A complex encompassing PPP2CA, PPP2R2A, and PPP2R1A was immunopurified from 293T cells transfected with pFlag-CMV2-PPP2R2A, pFlag-CMV2-PPP2R1A, and pcDNA3-HA-PPP2CA. Overall, 0.5 M NaCl was used in the washing buffer to remove PP2A-associated proteins. Flag-BRAF proteins were purified as described above in the In vitro kinase assays section. PP2A activity was assayed by incubating the immunopurified PP2A proteins with Flag-BRAF in the phosphatase assay buffer (25 mM Tris-HCl, pH 7.5, 1 mM EDTA, 1 mM EGTA, 1 mM dithiothreitol,

and 0.25 mg·ml$^{-1}$ bovine serum albumin) at 30 °C for 60 min prior to detection with malachite green phosphate detection solution (Promega)[55].

**In vivo ubiquitination assay.** In total, 293T cells were transfected with Flag-BRAF, His-ubiquitin, and Myc-ITCH. Thirty-six hours after transfection, 10 µM MG132 was added to block proteasome degradation, and cells were harvested in denatured buffer (6 M guanidine-HCl, 0.1 M Na$_2$HPO$_4$/NaH$_2$PO$_4$, and 10 mM imidazole), followed by Ni-NTA (Ni-nitrilotriacetic acid) purification and immunoblot analysis.

**In vitro ubiquitination assay.** GST-WT-ITCH, GST-C832S-ITCH, and His-BRAF proteins were expressed and purified in BL21 (DE3) *E. coli*. The ITCH in vitro ubiquitination assay was performed in a 30 -µl ubiquitination assay buffer (20 mM HEPES, pH 7.2, 5 mM MgCl$_2$, 0.1 mM DTT, and 1 mM ATP), with 50 nM E1, 500 nM UBCH7 (E2), and 1 µg of ubiquitin or ubiquitin variants (Boston Biochem). In total, 500 ng of GST-ITCH (wild type or mutant), together with 500 ng of His-BRAF or immunopurified Flag-BRAF proteins, were added to the buffer to initiate the reaction. Samples were incubated at 30 °C for 60 min. The reactions were stopped by the addition of 2 × SDS-PAGE sample buffer, and the reaction products were resolved by SDS-PAGE and probed with the indicated antibodies[32,56].

**Mass spectrometry analysis.** Flag-BRAF and HA-UB constructs were transfected into 293T cells. Forty-eight hours post transfection, cells were lysed for anti-Flag IP. The immunopurified Flag-BRAF was eluted using Flag peptide (Sigma-Aldrich) and subjected to anti-HA IP to enrich ubiquitinated BRAF proteins. Bound proteins were thermally denatured for 5 min at 95 °C, and then treated with DTT and iodoacetamide to reduce and alkylate proteins, followed by trypsin digestion. The solution containing tryptic peptides was acidified, and peptides were desalted using ZipTip-C18 (Millipore) pipette tip columns; the final elution was dried using vacuum centrifugation and resuspended in aqueous 2% acetonitrile and 0.1% formic acid. A nanoflow liquid chromatography (UHPLC Ultimate 3000, Dionex, Sunnyvale, CA) interfaced with a nanoelectrospray ion source online to a mass spectrometer (Q-Exactive Plus, Thermo Fisher Scientific) was used for tandem mass spectrometry peptide-sequencing experiments. The sample was first loaded onto a pre-column (2 cm × 100 µm ID packed with C18 reversed-phase resin, 5 µm, 100 Å, Thermo Scientific) and washed for 8 min at 4 µl per minute with aqueous solution, 2% acetonitrile, and 0.04% trifluoroacetic acid. The trapped peptides were eluted onto the analytical column (C18, 75 µm ID × 25 cm, Acclaim PepMap 100, Thermo Scientific). The 90-min run was programmed with solvent A (2% acetonitrile + 0.1% formic acid) and solvent B (90% acetonitrile + 0.1% formic acid) for 8 min, solvent B from 2 to 30% in 55 min, and then solvent B from 30% to 38.5% B in 5 min, ramped up to 90% B in 2 min, and held at 90% for 3 min, followed by solvent B from 90 to 2% in 2 min and final re-equilibration at 98% A solvent for 15 min. The flow rate on the analytical column was 300 nl per minute. The top 16 tandem mass spectra were collected in a data-dependent acquisition manner, following each survey scan. Data collected were analyzed in Proteome Discoverer Software (Thermo Fisher Scientific Inc.)

**Circular dichroism spectroscopy.** Circular dichroism (CD) spectra were measured on an Aviv 215 circular dichroism spectrometer using a 1 -mm path length quartz cuvette. Protein was diluted by Tris-NaCl buffer (pH 7.5). Three independent experiments were carried out and the spectra were averaged. The final spectra were normalized by subtracting the average blank spectra. Molar ellipticity [θ] (in deg·cm$^2$·dmol$^{-1}$) was calculated using the equation $[\theta] = 100\theta_{obs}(Cnl)^{-1}$, where $\theta_{obs}$ is the measured ellipticity in millidegrees, $n$ is the number of amino acid residues, $l$ is the path length in centimeters, and $C$ is the concentration of protein in millimoles.

**Clonogenic survival and soft agar assays.** The clonogenic survival and soft agar assays for WM1346, WM3918, melan-a, M245, and IPC-298 cells were performed by culturing cells in 10% FBS containing RPMI-1640 before plating into a six-well plate at 1000 cells per well. Two weeks later, cells were stained with crystal violet and the colony numbers were counted.

For soft agar assays, cells (10,000 per well) were seeded in 0.5% low-melting-point agarose in RPMI-1640 with 10% FBS, layered onto 0.8% agarose in RPMI-1640 with 10% FBS. The plates were kept in the cell culture incubator for 21 days, after which the colonies > 50 µm were counted under a light microscope. Formed colonies were stained with iodonitrotetrazolium chloride (INT).

**Matrigel invasion assay.** Melanoma cells were overlaid onto Transwell inserts coated with Matrigel (BD Biosciences) and allowed to invade for 24 h. Cells were then fixed and stained with phalloidin-AF594 and noninvasive cells were removed before fluorescence imaging with an inverted Nikon Eclipse TS100 microscope. To quantify the levels of invasion, fixed and stained cells were imaged with a Zeiss confocal microscope (20×) at 0 µm with 0.5-µm image slices taken throughout the distance of invasion.

**Transendothelial cell migration assays**. Transendothelial cell migration assays were performed by plating human umbilical vein endothelial cells (HUVEC) into Transwell inserts and grown to confluence. DiI-labeled melanoma cell lines were plated on top of the HUVEC layer and allowed to invade for 4 h. Nonmigratory cells were removed before imaging with an inverted Nikon Eclipse TS100 microscope. Calculations of the percentage of melanoma cells migrated through the confluent endothelial cell monolayers were performed with ImageJ analysis software[57].

**THP1 differentiation and transwell coculture assay**. THP1 cells were differentiated using Transwell inserts (BD Biosciences). To differentiate THP1 cells into M2-activated macrophages, THP1 cells were treated with $10 \, ng \cdot mL^{-1}$ of 12-O-tetradecanoylphorbol-13-acetate (TPA) for 24 h and subsequently with $20 \, ng \cdot mL^{-1}$ of IL4 and $20 \, ng \cdot mL^{-1}$ of IL13 for 48 h. After differentiation, the inserts were washed in RPMI three times before being placed into wells with pre-plated melanoma cells. Cell growth was monitored for up to 5 days before crystal violet staining[6].

**Xenograft tumor growth**. For assaying tumor growth in the xenograft model, 6-week-old male nude mice (NCRNU-M-M from Taconic) housed in specific pathogen-free environments were injected subcutaneously with $1.0 \times 10^6$ WM1346 derivatives ($n = 6$ for each group) mixed with the serum-free RPMI-1640 medium. Tumor size was measured every other day with a caliper, and the tumor volume was determined by the formula: $L \times W^2 \times 0.5$, where $L$ is the longest diameter and $W$ is the shortest diameter. Twenty-one days after injection, mice were euthanized and xenograft solid tumors were dissected. For experiments using doxycycline-induced expression of shRNA or cDNA, $625 \, mg \cdot kg^{-1}$ doxycycline-containing diet (Envigo) was administered to the animal for 14 days. All experimental procedures strictly complied with the IACUC guidelines and were approved by the IACUC of the University of South Florida.

**Statistical analyses**. All quantitative data were presented as the mean ± SEM (standard error of the mean) or the mean ± SD (standard deviation), as indicated of at least three independent experiments by Student's $t$ test for between-group differences. The $P < 0.05$ was considered statistically significant.

**Reporting summary**. Further information on research design is available in the Nature Research Reporting Summary linked to this article.

## Data availability
Full scans of the gels and blots are available in Supplementary Fig. 12. All relevant data are available from the corresponding author upon reasonable request.

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

## Acknowledgements

We thank Roger J. Davis, Edward W. Harhaj, Hans R. Widlund, Kay Hänggi, and Brian Ruffell for providing reagents; Florian A. Karreth, Eric K. Lau, Peter A. Forsyth, and members of the Wan and Smalley labs for useful discussions. This work was supported in part by the NIH grants (L.W., CA183914), the Skin SPORE (CA168536) Developmental Research Program, the Melanoma Research Alliance, and Forma Therapeutics, Inc. This work has been supported in part by the Proteomics and Metabolomics Core Facility at the H. Lee Moffitt Cancer Center & Research Institute, an NCI designated Comprehensive Cancer Center (P30-CA076292).

## Author contributions

Conceptualization: Q.Y. and L.W.; methodology: Q.Y., L.W., B.F., E.R.R., M.Z., C.Z., G.Z. and V.I.; investigation: Q.Y., T.H., S.J., E.R.R., V.I., C.Z., X.Y. and L.W.; writing—original draft: Q.Y. and L.W.; writing—review and editing: Q.Y. and L.W.; funding acquisition: L.W., J.C., E.B.H., M.K., J.M.K. and K.S.M.S.; resources: B.F., E.R.R., V.I. and J.M.K.; supervision: B.F., J.C., E.B.H., M.K., J.M.K., K.S.M.S. and L.W.

## Additional information

**Competing interests:** The authors declare no competing interests.

