## [Peer Review File · Nature Communications]

Reviewers' comments:

Reviewer #1, Expertise: E3 ligase, ubiquitination, immune regulation (Remarks to the Author):

In this study, the authors show that ITCH-mediated non-degradable ubiquitination of BRAF via K27-linked poly-ubiquitin chain in response to pro-inflammatory cytokines from tumor microenvironment in melanoma. K27-linked ubiquitination of BRAF controls ability to bind with inhibitory 14-3-3, leading to activate MEK/ERK signaling and melanoma cell survival. This study provides scientific insights for atypical ubiquitin mediated post translational modification of BRAF by ITCH in the regulation of melanoma survival. Although the data are in general convincing, one major concern is that whether such observations will have biological support, namely, whether it is relevant from the point view of Itch-deficient mice. Although the authors used Itch^{-/-} MEFs, how this would be related to BRAF-controlled transformation in melanoma remains unclear. Since Itch-promoted K27-chain formation is already known, the significance of findings in this paper is quite limited at the current stage.

there are other issues that are listed below:

1. Page 7, line 145 : 'K7-linked' should be 'K27-linked'.
2. ITCH activity and ITCH-mediated K27 ubiquitination of BRAF should be compared in normal skin cells and melanoma cells to validate physiological importance of ITCH-mediated K27 ubiquitination of BRAF for survival and invasion of melanoma.
3. In Fig 3b, the data show that the expression level of ITCH protein decrease at 20 min and 30 min after TNF α treatment. Reduced ITCH expression and increased p-MEK/p-ERK expressions appears contradictory. The authors need to address the possible mechanism that TNF α affects stability of ITCH and to explain whether the decreased ITCH could maintain the ability to activate BRAF/MEK/ERK signaling pathway.
4. In Fig 4a and 4b, the authors showed the effect of TNF α on MEK/ERK pathway activation accompanied with the activation of JNK. The protein expression of ITCH and direct evidence of ITCH activity (phosphorylated ITCH, c-FLIP...) should be tested to support the coordination between BRAF/MEK/ERK and JNK/ITCH signaling in response to TNF α .
5. In Fig 4c, to clarify whether TNF α induced ubiquitination of BRAF is ITCH-dependent, ubiquitination assay should be conducted in ITCH $+/+$ and ITCH $-/-$ cells with or without TNF α treatment.
6. From Fig 3 to 6, although melanoma cell lines were treated with TNF α or co-cultured with M2 differentiated THP1, these experiments are not enough to use terms of 'cytokine response'. The authors need to identify cytokines from M2 THP1 macrophages or tumor microenvironments and to provide additional data using identified cytokines as well as TNF α .
7. Page11, line 246 "Cytokine Recruits....." should be "TNF α recruit....".
8. In Fig6, the data are not enough to insist the notion that PPP2R2A has ability to dephosphorylate not p-S729 but p-S365. Additional experiments to measure dephosphorylate kinetics of PPP2R2A are needed.
9. Because there are important differences in phenotype and physiological responses between immortalized tumor cell line and primary tumor cells from the patient, the authors need to conduct the key experiments using primary melanoma cells.

10. The reference of 63 (Fang) should cite the original NI paper.

Reviewer #2, Expertise : BRAF, melanoma, Raf/Ras signalling (Remarks to the Author):

The manuscript by Yin et al presents data implicating K27-linked ubiquitination of wild-type BRAF as a mechanism that modulates BRAF (and presumably also CRAF and ARAF)-mediated ERK pathway activation. The authors use various biochemical and cell based assays to support a model by which ubiquitination of BRAF is mediated by the ITCH E3 ligase via K27, leading to release of inhibitory interaction with 14-3-3 and activation of MEK/ERK signaling in the context of cytokine stimulation. The study presents interesting findings with potentially important implications, but it is somewhat uneven. Part of the manuscript shows convincing evidence of BRAF ubiquitination, the role of ITCH and the effect of ubiquitination on BRAF's ability to activate ERK in cells. However, the molecular details of the proposed mechanism of BRAF activation by ubiquitination are less well supported by the data, and certain conclusions appear somewhat self-contradictory.

Specific comments:

1. Fig. 1, 2. In ex vivo assays for ubiquitination (co-IPs), the authors use proteasome inhibitor MG132 for 12 hours. Since, as they mention, K27-linked ubiquitination is non-degradative would the results be similar without proteasome inhibition?
2. Fig. 3. The authors' suggestion that the effect of ubiquitination on BRAF's ability to activate ERK is through the JNK/ITCH axis is based mostly on indirect evidence. Only TNF α is used to stimulate ERK activity in MEFs. Have they tried to stimulate ERK with PDGF, a known activator of ERK signaling in MEFs? Would the effect of PDGF on ERK also be hampered in Itch-/- MEFs? In such case, alternative mechanisms to JNK should be considered. For example, is the interaction of BRAF with MEK in cells affected by ubiquitination?
3. Fig 4. As in previous comment, only TNF α is used in RAS-mutant cell lines. What would be the effect of growth factors, such as EGF for example, on BRAF ubiquitination?
4. Fig 5. It is not clear how apparent BRAF activity in cells (as measured by ERK phosphorylation) is reduced in 5KR but the in vitro kinase activity is not affected. If, as the authors suggest, K-27-linked ubiquitination negatively regulates S365 phosphorylation in BRAF, then the in vitro activity of BRAF should be affected too (see Guan KL et al., JBC, 2000).
5. Fig 6. As in previous comment, data showing S365 and S729 phosphorylation status of 5KR, S675A/P676A and P751A/P754A BRAF are needed in support of the proposed model. If correct, phospho-S365 should be increased compared to wild-type BRAF, whereas S729 phosphorylation should be the same as wild-type.
6. The in vivo experiments in Fig 7 would be more relevant if knockdown and overexpression were inducible, so that the authors could study the effect of overexpression of 5KR-BRAF in established tumors.
7. In Supp. Figure 1e, there are significantly higher levels of ubiquitination of BRAF(V600E) compared to WT. How does this relate to the overall model the authors propose?

Point-by-Point Responses to the Reviewers' Critiques (NCOMMS-18-20081)

Reviewer #1, Expertise: E3 ligase, ubiquitination, immune regulation (Remarks to the Author):

In this study, the authors show that ITCH-mediated non-degradable ubiquitination of BRAF via K27-linked poly-ubiquitin chain in response to pro-inflammatory cytokines from tumor microenvironment in melanoma. K27-linked ubiquitination of BRAF controls ability to bind with inhibitory 14-3-3, leading to activate MEK/ERK signaling and melanoma cell survival. This study provides scientific insights for atypical ubiquitin mediated post translational modification of BRAF by ITCH in the regulation of melanoma survival. Although the data are in general convincing, one major concern is that whether such observations will have biological support, namely, whether it is relevant from the point view of Itch-deficient mice. Although the authors used Itch-/- MEFs, how this would be related to BRAF-controlled transformation in melanoma remains unclear. Since Itch-promoted K27-chain formation is already known, the significance of findings in this paper is quite limited at the current stage.

Response: We thank the reviewer for recognizing the novelty and the potentially impact of this study. We also agree with the reviewer that it is important to provide further experimental evidence to demonstrate that ITCH-mediated atypical ubiquitination of BRAF is an important mechanism to modulate the MAPK pathway, particularly in the setting of cytokine/TNF α treatment. Enlightened by the constructive comments from the reviewer, we have obtained the following results to substantiate the physiological importance of our findings.

- 1) In support of a positive role of ITCH in activating the RAF/MEK/ERK signaling cascade, we found that ectopic expression of WT-ITCH or a constitutive active form of ITCH (3D-ITCH) in murine melanocyte melan-a led to elevated MEK and ERK activities (**Fig. 4m**). The 3D-ITCH mutant (S199D, T222D and S232D) was generated to mimic the three JNK phosphorylation sites that serve to activate ITCH (*Chang L et al. (2006). Cell. 124, 601–613*).
- 2) More importantly, overexpression of ITCH supported TPA-independent growth of melan-a cells (**Fig. 4n-o**). Since proliferation of melan-a cells *in vitro* require growth factors or mitogens such as TPA (12-O-Tetradecanoylphorbol-13-acetate), TPA-independent growth has been used in multiple studies as an evidence for melanocyte transformation (*Garraway LA et al. (2005). Nature. 436, 117–122; Wan L et al. (2017). Cancer Discovery. 7, 424–441*). Moreover, although expression of 3D-ITCH alone failed to promote anchorage-independent growth of melan-a cells (**Fig. 4p-q**), when *Pten* was further depleted (**Supplementary Fig. 7m-o**), 3D-ITCH-expressing melan-a cells formed colonies in the soft agar (**Fig. 4p-q**). These results together suggest an oncogenic function of ITCH in transforming melanocytes, which is partly through activating the RAF/MEK/ERK signaling pathway.
- 3) In addition to MAPK activation, increased MITF expression was also observed in ITCH-overexpressed melanocytes (**Fig. 4m**). The upregulated MITF level may be due to decreased c-Jun expression as previous reports demonstrated that MITF is negatively regulated by c-Jun in melanoma cells (*Riesenberg S et al. (2015). Nature Communications. 6, 8755; Smith MP et al. (2014). Cancer Discovery. 4, 1214–1229*). In further support of this notion, depletion of ITCH in melanoma cells resulted in a decrease of MITF (**Fig. 3c-d**). MITF has been shown to promote melanocyte and melanoma proliferation (*Goding CR (2000). Genes and Development. 14, 1712–1728; Garraway LA et al. (2005). Nature. 436, 117–122*), the positive regulation of MITF by ITCH in melanoma cells may function as another mechanism through which ITCH facilitates melanocyte transformation.
- 4) To further interrogate an oncogenic role for ITCH in promoting melanoma cell survival, we generated a WM1346 melanoma cell line expressing a doxycycline-inducible sh*ITCH* construct

(backbone EZ-Tet-pLKO-Hygro from Addgene). We found a marked decrease of pMEK and pERK levels as well as cell growth after doxycycline treatment (**Fig. 3e** and **3j**). Furthermore, doxycycline treatment *in vivo* significantly suppressed the growth of Teton-shITCH-WM1346 melanoma cell in a xenograft mouse model (**Fig. 3l-n**).

- 5) Different from the *Itch*^{a18H/a18H} mouse, which is viable (Perry WL *et al.* (1998). *Nature Genetics* 18, 143–146; Fang D *et al.* (2002). *Nature Immunology*. 3, 281–287), germline deletion of *Braf* tended to be embryonic lethal due to vascular defects (Wojnowski L *et al.* (1997). *Nature Genetics* 16, 293–297). We compared the cell viability and growth of melan-a cells with shRNAs targeting *Itch* or *Braf*, and found that melan-a cells were quite tolerate to either *Itch* or *Braf* deficiency (**Supplementary Fig. 5e-g**). Moreover, depletion of *Braf* resulted in a slightly enhanced growth inhibition compared to the *Itch* knockdown (**Supplementary Fig. 5e-g**). On the contrary, in melanoma cells, depletion of either *ITCH* or *BRAF* resulted in a massive cell growth arrest and subsequent cell death (**Fig. 3i-j** and **Supplementary Fig. 4j-p**). These findings support the notion that melanoma cells are more addicted to the RAS/RAF/MEK/ERK signaling cascade to support cell survival. The molecular modules including *ITCH* that fuels MAPK hyperactivation are therefore vital to melanoma cells. However, in normal cells where minimal MAPK signaling strength is required, *ITCH*-mediated *BRAF* activation might be relatively dispensable. We agree with the reviewer that an *Itch*-deficient melanomagenesis mouse model will be critical to further assess the pathophysiological role of *ITCH* in melanoma development. However, due to the large volume of data associated with this manuscript, we hope the reviewer could concur with us that these studies would be more suitable for a separate manuscript in the near future.

There are other issues that are listed below:

1. Page 7, line 145 : *~K7-linked~™ should be ~K27-linked~™.*

Response: We apologize for this mistake; the correction has been made in the revised manuscript.

2. *ITCH activity and ITCH-mediated K27 ubiquitination of BRAF should be compared in normal skin cells and melanoma cells to validate physiological importance of ITCH-mediated K27 ubiquitination of BRAF for survival and invasion of melanoma.*

Response: We agree with the reviewer that it is critical to compare the *ITCH*-mediated K27 ubiquitination of *BRAF* in both normal melanocytes and melanoma cells. As kindly instructed, in the **Supplementary Fig. 6e**, we found that TNF α stimulated K27-linked ubiquitination of *BRAF* in both melan-a and WM1346 cells. Compared to melan-a, TNF α -stimulated *BRAF* ubiquitination appeared slightly stronger in WM1346 cells. Moreover, depleting *ITCH* in both cells led to a marked reduction of *BRAF* ubiquitination upon TNF α treatment (**Supplementary Fig. 6i-j**), suggesting that in both normal melanocytes and melanoma cells, *ITCH* promotes *BRAF* ubiquitination in response to cytokine stimulation. As we have addressed above, compared to melanoma cells (**Fig. 3i-j**), depletion of *Itch* in melanocytes only moderately suppressed melan-a cells proliferation (**Supplementary Fig. 5e-g**). These results indicate that although *ITCH*-mediated *BRAF* K27-ubiquitination occurs in both cell types, due to the oncogenic addiction of melanoma cells to the MAPK signaling, the loss of *ITCH*, which significantly abolishes MEK and ERK activity in melanoma cells, led to a much stronger inhibition of cell proliferation in melanoma cells (**Fig. 3i-n**). These results thus suggest a therapeutic window for targeting *ITCH* in melanoma patients, a future direction warrants in-depth investigation.

3. *In Fig 3b, the data show that the expression level of ITCH protein decrease at 20 min and 30 min after TNF α treatment. Reduced ITCH expression and increased p-MEK/p-ERK expressions appears*

contradictory. The authors need to address the possible mechanism that TNF α affects stability of ITCH and to explain whether the decreased ITCH could maintain the ability to activate BRAF/MEK/ERK signaling pathway.

Response: We thank the reviewer for pointing out that in the original Fig. 3b, 20 min after TNF α treatment, Itch protein level was reduced. We also agree with the reviewer that it is important to examine if TNF α treatment affects ITCH stability/activity in both MEFs and melanoma cells. As kindly suggested, in the revised manuscript, we have obtained the following experimental evidence to address this concern:

- 1) ITCH stability is mainly controlled by its auto-ubiquitination and subsequent degradation. We found that compared to WT-ITCH, which exhibited an approximate four-hour half-life (**Supplementary Fig. 5a-b**), the catalytically inactive CS-ITCH was completely stabilized (**Supplementary Fig. 5b**). In support of this notion, c-Jun was significantly stabilized in CS-ITCH-expressing cells as evidenced by an extended half-life (**Supplementary Fig. 5b**).
- 2) TNF α treatment didn't significantly destabilize ITCH as we found in melanoma cells, 240 min after TNF α or IL-1 β treatment, the ITCH protein level remained largely unchanged (**Fig. 4a-b** and **Supplementary Fig. 6a-d**). We also rerun the lysates of original Fig. 3b, and found that the previous reduction of ITCH at 20 and 30 min was likely due to the reblotting issue. In the revised **Fig. 3b**, the ITCH protein level remains unchanged at the 20 and 30 min time points.
- 3) Since in our experimental conditions such as in **Fig. 3b** and **Fig. 4a-b**, newly synthesized ITCH may complement the loss of ITCH caused by proteolysis. Following the reviewer's insightful comment, we compared the half-lives of the ITCH protein in melanoma cells upon TNF α treatment. As shown in **Supplementary Fig. 5c**, TNF α appears to destabilize ITCH likely through accelerating its auto-ubiquitination since the CS-ITCH remained stable even in the presence of TNF α .
- 4) Furthermore, in support of a non-proteolytic function of ITCH-mediated BRAF ubiquitination, BRAF stability was not affected in the experimental conditions we examined (**Supplementary Fig. 5a-c**).

Taken together, our newly obtained results suggest that TNF α activates ITCH and promotes its auto-ubiquitination and degradation, while at the same time promotes ITCH-mediated ubiquitination of substrates including BRAF and c-Jun (**Supplementary Fig. 7c**), which in turn activates BRAF and destabilizes c-Jun, respectively. The simultaneous activating an ubiquitin E3 ligase and destabilizing it via auto-ubiquitination has been observed for a number of E3 ligases including ITCH and other NEDD4 family E3 ligases (Gallagher *E et al.* (2006). *Proc Natl Acad Sci U S A.* 103, 1717–1722; Chen *Z et al.* (2017). *Mol Cell.* 66, 342–345).

4. In Fig 4a and 4b, the authors showed the effect of TNF α on MEK/ERK pathway activation accompanied with the activation of JNK. The protein expression of ITCH and direct evidence of ITCH activity (phosphorylated ITCH, c-FLIP β) should be tested to support the coordination between BRAF/MEK/ERK and JNK/ITCH signaling in response to TNF α .

Response: We thank the reviewer for this brilliant suggestion to examine ITCH phosphorylation and subsequent activation after TNF α treatment. As kindly instructed, we found that TNF α treatment led to a marked increase of ITCH serine/threonine phosphorylation using the p-(S/T)P antibody against immunopurified ITCH protein (**Supplementary Fig. 7a**). Furthermore, as described in the response to comment #3, we found that both ITCH and its ubiquitin substrate c-Jun were destabilized after TNF α treatment as evidenced by shortened half-lives, whereas in cells expressing catalytic inactive CS-ITCH,

both ITCH and c-Jun were stabilized (**Supplementary Fig. 5b-c**). Our newly obtained results thus support previous reports that JNK phosphorylation relieves the auto-inhibitory conformation of the ITCH protein (Gallagher E et al. (2006). *Proc Natl Acad Sci U S A.* 103, 1717–1722), which in turn promotes the poly-ubiquitination of its downstream targets including proteolytic substrates c-Jun, JunB and c-FLIP as well as non-proteolytic substrates like BRAF.

To provide additional evidence to support this notion, we generated WM1346-sh*ITCH* cell lines stably expressing WT-ITCH or JNK phosphorylation deficient 3A (S199A/T222A/S232A)-ITCH. As shown in **Supplementary Fig. 7c**, compared to WT-ITCH-expressing cells, TNF α -induced poly-ubiquitination of both BRAF and c-Jun were largely abolished. Furthermore, 3A-ITCH-expressing WM1346 cells were blunted to TNF α -triggered MEK/ERK activation (**Supplementary Fig. 7d**). Hence, our results demonstrate a clear link between TNF α -mediated JNK/ITCH activation and subsequent BRAF/MEK/ERK activation.

We also analyzed the TCGA cutaneous melanoma RPPA (reverse phase protein array) dataset (TCGA-SKCM) with 354 tumor samples (Li J et al. (2013). *Nature Methods.* 10, 1046–1047). Notably, we found a positive correlation ($r=0.28$, $p=5.4\times 10^{-8}$) between pT183/pY185-JNK1/2 and pT202/pY204-ERK1/2 signals from this cohort of studies (**Supplementary Fig. 7b**). This piece of proteomic information from melanoma tumor samples provided a footnote to further support our conclusion.

5. In Fig 4c, to clarify whether TNF α induced ubiquitination of BRAF is ITCH-dependent, ubiquitination assay should be conducted in ITCH $+/+$ and ITCH $-/-$ cells with or without TNF α treatment.

Response: As kindly suggested by the reviewer, in **Fig. 4e**, we demonstrated that there was a higher BRAF poly-ubiquitination in *Itch* $^{+/+}$ MEFs compared to *Itch* $^{-/-}$ MEFs. Further, TNF α treatment promoted BRAF poly-ubiquitination only in *Itch* $^{+/+}$ MEFs but not its null counterparts. Further, TNF α was not capable of promoting BRAF ubiquitination in *ITCH*-depleted melanoma cells or WM1346 melanoma cells (**Supplementary Fig. 6i-j**). In concert with the results from MEFs (**Fig. 3b**), we found that MEK/ERK activation is largely refractory to TNF α treatment in melanoma cells depleted of *ITCH* (**Fig. 4d** and **Supplementary Fig. 6p**). These results again support the correlation between TNF α -mediated JNK/ITCH activation and subsequent BRAF/MEK/ERK activation.

6. From Fig 3 to 6, although melanoma cell lines were treated with TNF α or co-cultured with M2 differentiated THP1, these experiments are not enough to use terms of “cytokine response”. The authors need to identify cytokines from M2 THP1 macrophages or tumor microenvironments and to provide additional data using identified cytokines as well as TNF α .

Response: We agree with the reviewer that it is not accurate to use the term “cytokine response” in our original manuscript given our results only used TNF α to stimulate the JNK/ITCH pathway in our experimental settings. Previous studies from the Wellbrock group (Smith MP et al. (2014). *Cancer Discovery.* 4, 1214–1229) demonstrated that TNF α from melanoma tumor-associated macrophages facilitates tumor growth and contributes to BRAF inhibition resistance. The authors revealed that TNF α is highly enriched in M1- and M2-THP1-conditioned media, and that TNF α treatment stimulated the proliferation of melanoma cell line WM266.4. To demonstrate that TNF α is also a key factor to promote melanoma cell growth in our experimental settings, we adopted the approach used in the Smith et al. paper by using TNF α -blocking antibody in our M2 THP1-melanoma cell co-culture experiment. As shown in **Supplementary Fig. 7k-l**, we found that the antibody against TNF α suppressed THP1-stimulated WM3918 melanoma cell growth while had no effect on the proliferation of WM3918 cells in

the absence of M2 THP1. This finding also suggests that TNF α is the major factor from M2-THP1 to facilitate melanoma cell growth.

The major conclusion of our manuscript is that the JNK/ITCH signaling pathway facilitates MEK/ERK activation via atypical ubiquitination of BRAF. We fully agree with the reviewer that in addition to TNF α , there might be other cytokines that could also activate the JNK/ITCH pathway to fuel the MEK/ERK signaling in melanoma cells. For example, CAFs (cancer-associated fibroblasts) exhibit tumor-promoting function through secreting pro-inflammatory cytokines (Erez *N et al. (2010). Cancer Cell. 17, 135–147*). Among CAFs-derived cytokines, IL-1 β promotes melanoma invasion and angiogenesis (Voronov *E et al. (2003). Proc Natl Acad Sci U S A. 100, 2645–2650*). Enlightened from the reviewer's constructive comment, we found that analogous to TNF α , IL-1 β activated JNK and stimulated pMEK and pERK in melanoma cells (**Supplementary Fig. 6c-d**). Furthermore, IL-1 β promoted the growth of melanoma cells in clonogenic survival experiments (**Supplementary Fig. 7i-j**). These results coherently support the notion that the JNK/ITCH signaling pathway engages upstream stimulations including TNF α and IL-1 β to promote BRAF poly-ubiquitination and subsequent MEK/ERK activation (**Supplementary Fig. 11m**).

7. Page11, line 246 *“Cytokine Recruits”* should be *“TNF α recruit”*.

Response: We fully agree with reviewer that it is not accurate to use “Cytokine” in the original manuscript given that only TNF α was used in our experiments. In the revised manuscript, following the constructive comments from the reviewer, experimental results using IL-1 β were included to demonstrate that similar to TNF α , IL-1 β activated the JNK/ITCH module to accelerate BRAF poly-ubiquitination (**Supplementary Fig. 6h**) and to stimulate BRAF/MEK/ERK in melanoma cells (**Supplementary Fig. 6c-d**), which in turn facilitated proliferation of melanoma cells (**Supplementary Fig. 7i-j**). Therefore, we hope the reviewer would concur with us that it is now appropriate to use the term “Cytokine” in the subtitle for Fig. 4.

8. In Fig6, the data are not enough to insist the notion that PPP2R2A has ability to dephosphorylate not p-S729 but p-S365. Additional experiments to measure dephosphorylate kinetics of PPP2R2A are needed.

Response: We completely agree with the reviewer that additional experiments to support a role of PPP2R2A in recruiting the PP2A complex to the ubiquitination BRAF are required. To this end, in the revised manuscript, we have obtained the following experimental results to further demonstrate that PPP2R2A facilitates PP2A-mediated dephosphorylation of BRAF p-S365:

- 1) Depletion of the regulatory subunit of PP2A, PPP2R2A, led to an increase of pS365-BRAF, but not pS729-BRAF (**Fig. 6l** and **Supplementary Fig. 9m**). Both PPP2CA and PPP2R2A knockdown led to reduced pMEK and pERK levels (**Fig. 6k-l** and **Supplementary Fig. 9m**). These results indicate that the PPP2R2A-containing PP2A complex mainly targets pS365-BRAF rather than pS729-BRAF. Interestingly, previous studies revealed that PP2A also primarily dephosphorylates pS259-CRAF but not pS621-CRAF, equivalent sites to pS365 and pS729-BRAF (Abraham *D et al. (2000). J. Biol. Chem. 275, 22300–22304*; Jaumot *M & Hancock JF (2001). Oncogene. 20, 3949–3958*; Ory *S et al. (2003). Current Biology. 13, 1356–1364*).
- 2) Moreover, PPP2R2A-depleted melanoma cells displayed a blunted response to TNF α -induced pS365-BRAF dephosphorylation as well as MEK/ERK activation (**Fig. 6m** and **Supplementary Fig. 9n**).

- 3) As suggested by the reviewer, we found that compared to the PPP2CA immunoprecipitates, the PP2A complex containing PPP2CA, PPP2R2A and PPP2R1A exhibited a higher activity in dephosphorylating pS365-BRAF but not pS729-BRAF (**Fig. 6h-i** and **Supplementary Fig. 9l**).
- 4) Further kinetic analyses revealed that the Michaelis–Menten kinetics of dephosphorylating BRAF proteins for PPP2CA immunoprecipitates was $V_{max}=4.21\pm0.63$ pmol/min/ μ g; $K_m=0.185\pm0.14$ μ M, while for the PP2A complex containing PPP2CA, PPP2R2A and PPP2R1A was $V_{max}=10.33\pm1.31$ pmol/min/ μ g; $K_m=0.74\pm0.39$ μ M (**Fig. 6i**). This result demonstrates that compared to the PPP2CA catalytic subunit alone, the PP2A complex containing the PPP2R2A subunit exhibited a higher phosphatase activity in dephosphorylating BRAF.
- 5) More importantly, compared to PPP2CA immunoprecipitates, ubiquitinated BRAF appeared to be a better substrate for the PP2A complex containing PPP2CA, PPP2R2A and PPP2R1A (**Fig. 6j**), indicating that PPP2R2A binds to the K27-linked poly-ubiquitin chain and recruits the PP2A complex to accelerate pS365-BRAF dephosphorylation (**Fig. 6e**).
- 6) In addition, a recent proteomic study revealed PPP2R2A as a BRAF interacting protein in cells (*Diedrich B et al. (2017). The EMBO Journal. 36, 646–663*).

Taken together, our newly obtained results further demonstrate a crucial role for the WD40 repeats domain containing subunit PPP2R2A in promoting pS365-BRAF dephosphorylation by the PP2A complex, a mechanism that leads to BRAF activation upon its K27-linked ubiquitination by ITCH. It will be important to further assess if other B55 subunits, which also contain the WD40 repeats domain, function similarly as PPP2R2A, and whether B56 and other regulatory subunits could bind to ubiquitinated BRAF and dephosphorylate pS365-BRAF. However, we hope the reviewer could concur with us that these studies are outside of the main scope of current manuscript and will be more suitable for a separate manuscript in the near future.

9. Because there are important differences in phenotype and physiological responses between immortalized tumor cell line and primary tumor cells from the patient, the authors need to conduct the key experiments using primary melanoma cells.

Response: We thank the reviewer for this constructive suggestion to examine if ITCH-mediated BRAF ubiquitination promotes MEK/ERK activation and melanoma proliferation. However, due to technical difficulty of propagating primary melanoma cells, especially those harboring WT-BRAF, for efficient *in vitro* shRNA knockdown experiments, we chose to use early passage (<20 passages) melanoma cell lines M245 and IPC-298 derived from melanoma patients obtained from Dr. Keiran Smalley lab for our experiments. These two cell lines both express WT-BRAF and mutant NRAS (**Supplementary Fig. 4d-e**). As shown in **Supplementary Fig. 4d-e**, depletion of ITCH in both M245 and IPC-298 cells led to reduced pMEK/pERK as we observed in other BRAF^{WT} melanoma cell lines. Notably, these cells displayed poor survival after ITCH knockdown as evidenced by clonogenic survival assays (**Supplementary Fig. 4n**). Further, TNF α treatment facilitates BRAF ubiquitination, MEK/ERK activation and cell proliferation in both cells (**Supplementary Fig. 6a-b, 6g** and **7g-h**). These results, together with our previous results using WM1346, WM3918 melanoma cell lines, normal melanocytes and MEFs coherently support our conclusion that ITCH-mediated BRAF ubiquitination activates BRAF activity to promote melanoma cell proliferation and invasion.

In addition to the results we obtained from our experimental settings. We analyzed the TCGA cutaneous melanoma RPPA (reverse phase protein array) dataset (TCGA-SKCM) with 354 tumor samples (*Li J et al. (2013). Nature Methods. 10, 1046–1047*). Interestingly, we found a positive correlation ($r=0.28$, $p=5.4\times10^{-8}$) between pT183/pY185-JNK1/2 and pT202/pY204-ERK1/2 signals from this cohort of studies (**Supplementary Fig. 7b**). This piece of proteomic information from melanoma tumor samples provided a footnote to further support our major conclusion.

10. *The reference of 63 (Fang) should cite the original NI paper.*

Response: As instructed, the original Nature Immunology paper (*Fang D et al. (2002). Nature Immunology. 3, 281–287*) has been cited in the revised manuscript.

Reviewer #2, Expertise: BRAF, melanoma, Raf/Ras signalling (Remarks to the Author):

The manuscript by Yin et al presents data implicating K27-linked ubiquitination of wild-type BRAF as a mechanism that modulates BRAF (and presumably also CRAF and ARAF)-mediated ERK pathway activation. The authors use various biochemical and cell based assays to support a model by which ubiquitination of BRAF is mediated by the ITCH E3 ligase via K27, leading to release of inhibitory interaction with 14-3-3 and activation of MEK/ERK signaling in the context of cytokine stimulation. The study presents interesting findings with potentially important implications, but it is somewhat uneven. Part of the manuscript shows convincing evidence of BRAF ubiquitination, the role of ITCH and the effect of ubiquitination on BRAF's ability to activate ERK in cells. However, the molecular details of the proposed mechanism of BRAF activation by ubiquitination are less well supported by the data, and certain conclusions appear somewhat self-contradictory.

Response: We thank the reviewer for recognizing the novelty of this study. We also agree with the reviewer that it is crucial to provide further experimental evidence to support the molecular mechanisms by which the kinase activity of poly-ubiquitinated BRAF is promoted. We hope after reading our point-by-point responses below as well as our revised manuscript, the reviewer would concur with us that we have obtained convincing results to demonstrate that our newly discovered molecular circuit serves to activate the RAF/MEK/ERK signaling cascade and contributed to melanoma cell survival.

Specific comments:

1. Fig. 1, 2. In ex vivo assays for ubiquitination (co-IPs), the authors use proteasome inhibitor MG132 for 12 hours. Since, as they mention, K27-linked ubiquitination is non-degradative would the results be similar without proteasome inhibition?

Response: We thank the reviewer for raising the concern regarding the use of MG132 in our ubiquitination assays. The reason that we use MG132 across the panels in Fig. 1 and Fig. 2 is to keep the experimental condition consistent for different ubiquitin mutants. As the reviewer kindly pointed out, wild type (WT) ubiquitin and the ubiquitin mutants that have K48 and K11 available could be utilized to assemble K48- or K11-linked poly-ubiquitin chains, which could be further processed by the 26S proteasome for substrate protein degradation. In our figure panels using WT-ubiquitin and ubiquitin mutants with K48 or K11 available, we added MG132 to the cells to ensure no K48/K11 linkages were disposed of from our ubiquitination assay results.

As kindly suggested by the reviewer, we performed the ubiquitination assay in **Fig. 1g** without MG132, as the reviewer could find out in the **Supplementary Fig. 1f**, the results are largely identical to the results we obtained from the condition with MG132.

*2. Fig. 3. The authors' suggestion that the effect of ubiquitination on BRAF's ability to activate ERK is through the JNK/ITCH axis is based mostly on indirect evidence. Only TNF α is used to stimulate ERK activity in MEFs. Have they tried to stimulate ERK with PDGF, a known activator of ERK signaling in MEFs? Would the effect of PDGF on ERK also be hampered in *Itch*^{-/-} MEFs? In such case, alternative mechanisms to JNK should be considered. For example, is the interaction of BRAF with MEK in cells affected by ubiquitination?*

Response: We thank the reviewer for the brilliant suggestion to further determine if in addition to TNF α , other cytokines or growth factors also activate BRAF via the JNK/ITCH module. In the original Supplementary Fig. 3a (revised **Supplementary Fig. 4a**), EGF was used to stimulate *Itch*^{+/+} and *Itch*^{-/-} MEFs and we found that EGF activated MEK and ERK regardless of *Itch* genetic status. However,

compared to *Itch*^{-/-} MEFs, *Itch*^{+/+} MEFs displayed a stronger and sustainable response to EGF (**Supplementary Fig. 4a**), indicating that although not exclusively, ITCH also participated in EGF-triggered MEK/ERK activation. Moreover, as kindly instructed by the reviewer, we compared the pMEK/pERK response of these MEFs to PDGF. As shown in **Supplementary Fig. 4b**, quite similar to EGF treatment, PDGF activated MEK and ERK in both *Itch*^{+/+} and *Itch*^{-/-} MEFs. On the other hand, for TNF α treatment, the MEK/ERK activation was almost completely abolished in *Itch*^{-/-} MEFs (**Fig. 3b**), suggesting an indispensable role of ITCH in mediating this process. From our results, we found that growth factors triggered a robust but transient activation of MEK/ERK, whereas cytokine-induced MEK/ERK was rather moderate but more sustained. Furthermore, depletion of *ITCH* in 293T cells (**Supplementary Fig. 1h**), melanocytes melan-a (**Supplementary Fig. 6i**) and WM1346 melanoma cells (**Supplementary Fig. 6j**) abrogated endogenous BRAF ubiquitination, again support a key role of ITCH in mediating TNF α -induced BRAF activation.

To provide additional evidence to support this notion, we generated WM1346-sh*ITCH* cell lines stably expressing WT-ITCH or JNK phosphorylation deficient 3A (S199A/T222A/S232A)-ITCH. As shown in **Supplementary Fig. 7c**, compared to WT-ITCH-expressing cells, TNF α -induced poly-ubiquitination of both BRAF and c-Jun were largely abolished. Furthermore, 3A-ITCH-expressing WM1346 cells were blunted to TNF α -triggered MEK/ERK activation (**Supplementary Fig. 7d**). Hence, our results demonstrate a clear link between TNF α -mediated JNK/ITCH activation and subsequent BRAF/MEK/ERK activation.

We also analyzed the TCGA cutaneous melanoma RPPA (reverse phase protein array) dataset (TCGA-SKCM) with 354 tumor samples (*Li J et al. (2013). Nature Methods. 10, 1046–1047*). Notably, we found a positive correlation ($r=0.28$, $p=5.4\times 10^{-8}$) between pT183/pY185-JNK1/2 and pT202/pY204-ERK1/2 signals from this cohort of studies (**Supplementary Fig. 7b**). This piece of proteomic information from melanoma tumor samples provided a footnote to further support our major conclusion.

Together, our results indicate that growth factors and TNF α utilize different signaling cascades to activate the RAF/MEK/ERK pathway. Our findings in this manuscript advocate a role for JNK/ITCH pathway to stimulate RAF proteins via ITCH-mediated poly-ubiquitination of RAF proteins. However, compared to the canonical growth factor/receptor tyrosine kinase (RTK)/RAS/RAF/MEK/ERK signaling cascade, the MEK/ERK activation is quite moderate after cytokine stimulation, which indicates that the JNK/ITCH axis is not the predominant signal to activate MEK/ERK in normal culturing conditions. However, for melanoma cells that expose to proinflammatory environment, a scenario that could be found in most solid tumors (*Lippitz BE (2013). The Lancet Oncology. 14, e218–e228*), the cytokine-mediated activation of MEK/ERK may provide a protection against cytokine-induced cytotoxicity. Previous studies have suggested a number of mechanisms to explain cytokine-triggered MEK/ERK activation (*Dumitru CD et al. (2000). Cell. 103, 1071–1083*; *Marques-Fernandez F et al. (2013). Cell Death and Disease. 4, e493*; *Shao Y et al. (2015). Journal of Investigative Dermatology. 135, 1839–1848*). Although the increase of c-FLIP transcription upon NF- κ B activation has been shown to regulate CRAF function in neuronal cells, the mechanism that leads to the immediate activation of ERK upon cytokine stimulation remains unclear, which was one of the reasons directed us to our current findings.

Enlightened by the comments from both reviewers, in addition to TNF α , we also treated melanoma cells with IL-1 β , another proinflammatory cytokine found in tumor microenvironment. As shown in the revised manuscript, IL-1 β treatment also led to the increase BRAF ubiquitination and activation of MEK/ERK (**Supplementary Fig. 6c-d and 6h**). Furthermore, similar to TNF α treatment, cells with the addition of IL-1 β displayed enhanced proliferation (**Supplementary Fig. 7i-j**).

We agree with reviewer that it is critical to examine if BRAF K27-linked poly-ubiquitination also influence its interaction with MEK, its direct downstream target. As shown in **Supplementary Fig.**

9a, *in vitro* ubiquitination of BRAF didn't affect its binding to MEK1. Moreover, TNF α treatment, which promoted BRAF ubiquitination in cells, failed to alter its interaction with MEK (**Fig. 6c** and **Supplementary Fig. 9g**). We have tested the binding of ubiquitinated BRAF with a number of its known upstream and downstream interacting proteins. Among all the proteins examined, only 14-3-3 but not MEK1, NRAS, BRAF, CRAF or KSR1, displayed reduced binding with ubiquitinated BRAF both *in vitro* (**Fig. 6a** and **Supplementary Fig. 9a-e**) and in cells (**Fig. 6c** and **Supplementary Fig. 9g**).

3. Fig 4. As in previous comment, only TNF α is used in RAS-mutant cell lines. What would be the effect of growth factors, such as EGF for example, on BRAF ubiquitination?

Response: As kindly suggested, in the revised manuscript, we examined how EGF might affect BRAF ubiquitination and downstream signals in melanoma cells. As shown in **Supplementary Fig. 6l**, similar to TNF α , EGF treatment promoted BRAF ubiquitination in WM1346 cells. Notably, in melanoma cells, EGF treatment also activated JNK (**Supplementary Fig. 6k**), which is likely through the EGFR/FAK/MEKK1/JNK signaling circuit (Huang C et al. (2004). *Journal of Cell Science*. 117, 4619–4628). To further interrogate whether EGF-induced BRAF ubiquitination is through ITCH, we compared BRAF ubiquitination in WT and *Itch*^{-/-} MEFs upon EGF treatment, as shown in **Supplementary Fig. 6m**, BRAF ubiquitination is largely hampered in *Itch*-deficient cells. These results together suggest that although EGF promotes BRAF ubiquitination through the JNK/ITCH axis, given the direct route from EGFR/RAS/RAF to ERK activation, EGF-dependent BRAF ubiquitination may only play a minor role in tuning BRAF function.

Moreover, enlightened by the constructive comment from the reviewer, we compared TNF α and EGF-stimulated BRAF ubiquitination in WT-BRAF and ubiquitination deficient 5KR-BRAF-expressing melanoma cells. As shown in the revised **Supplementary Fig. 10d-e**, we found that analogous to *ITCH* knockdown, 5KR-BRAF-expressing cells exhibited a diminished response to TNF α - and EGF-mediated BRAF ubiquitination. These results coherently support our conclusion that upstream signals that activate the JNK/ITCH signaling axis promote BRAF ubiquitination and subsequent MEK/ERK activation (**Supplementary Fig. 11m**).

In addition to JNK-mediated ITCH activation, the Ataxia Telangiectasia Mutated (ATM) kinase has been shown to catalyze ITCH S161 phosphorylation and to promote ITCH function in T cells (Santini S et al. (2014). *Oncogene*. 33, 1113–1123). Therefore it will be very intriguing to assess if ATM-mediated ITCH activation may play a role in modulating RAF/MEK/ERK signaling upon DNA damage, especially in the setting of UV irradiation, an important factor that contributes to melanomagenesis (Rass K & Reichrath J (2008). *Adv. Exp. Med. Biol.* 624, 162–178).

4. Fig 5. It is not clear how apparent BRAF activity in cells (as measured by ERK phosphorylation) is reduced in 5KR but the *in vitro* kinase activity is not affected. If, as the authors suggest, K-27-linked ubiquitination negatively regulates S365 phosphorylation in BRAF, then the *in vitro* activity of BRAF should be affected too (see Guan KL et al., *JBC*, 2000).

Response: We fully agree with the reviewer that it is puzzling that ubiquitination-deficient 5KR-BRAF displayed compromised activity in cells (**Fig. 5a**) while it exhibited full potential to phosphorylate GST-MEK1 in the *in vitro* kinase assay (**Fig. 5b**). To solve this discrepancy, first we compared the pS365 and pS729 levels of both WT-BRAF and 5KR-BRAF immunoprecipitates (IPs) that were used for *in vitro* kinase assays (**Supplementary Fig. 8e**). Surprisingly, both WT- and 5KR-BRAF displayed similar levels of pS365 and pS729. We used Flag-BRAF IPs from transiently transfected 293T cells for *in vitro* kinase assays. The exogenous Flag-BRAF expressed in 293T cells were apparently much more than the

endogenous BRAF (3 μ g Flag-BRAF was transfected to a 10 cm dish of 293T cells in order to prepare Flag-BRAF IPs). On the other hand, in assays to determine BRAF activity in cells, much less Flag-BRAF was transfected (90 ng BRAF was transfected to a 10 cm 293 cells). In the latter setting, the exogenous Flag-BRAF expressed at a level close to the endogenous BRAF level (**Supplementary Fig. 8f**). These results suggest that overexpressed Flag-BRAF in 293T cells are no longer modified in a fashion as the endogenous ones likely due to limited upstream kinases/phosphatases. This notion is supported by the observation that although lower doses of 5KR-BRAF (90 ng/dish) were hyperphosphorylated at S365 and thus incapable to activate MEK, high dose of 5KR-BRAF (1000 ng/dish) exhibited a comparable activity to its WT counterpart to promote MEK activation (**Supplementary Fig. 8f**). The relative band intensities were labelled underneath the blots, however, it is worth noting that the band intensities for the 1000 ng BRAF lanes were out of linear range, which could only be used as a close estimation.

We also thank the reviewer for mentioning previous studies by the Guan group that the inhibitory phosphorylation-deficient BRAF mutants displayed elevated activities *in vitro* (Guan KL *et al.* (2000). *J. Biol. Chem.* 275, 27354–27359; Chong H *et al.* (2001). *The EMBO Journal.* 20, 3716–3727; Zhang BH & Guan KL (2000). *The EMBO Journal.* 19, 5429–5439). Inspired by the insightful suggestion from the reviewer, HA-WT-BRAF and HA-5KR-BRAF proteins were immunopurified from WM1346 cells stably expressing exogenous BRAF at the endogenous level (**Fig. 7a-b**). Our results demonstrated that compared to WT-BRAF, 5KR-BRAF expressing cells exhibited lower MEK/ERK activity and were refractory to TNF α stimulation (**Fig. 7a-b**). Further *in vitro* kinase assays using HA-WT-BRAF and HA-5KR-BRAF IPs from WM1346 stable cell lines revealed that HA-WT-BRAF was more active than HA-5KR-BRAF in catalyzing GST-MEK1 phosphorylation (**Supplementary Fig. 8g**). Furthermore, HA-WT-BRAF purified from cells treated with TNF α displayed higher activity to promote GST-MEK1 phosphorylation, which might be due to its reduced pS365 inhibitory phosphorylation (**Supplementary Fig. 8g**). Together our newly obtained results elicited a model that ubiquitination of BRAF by ITCH mainly controls BRAF pS365 inhibitory phosphorylation rather than the intrinsic BRAF kinase function.

5. Fig 6. As in previous comment, data showing S365 and S729 phosphorylation status of 5KR, S675A/P676A and P751A/P754A BRAF are needed in support of the proposed model. If correct, phospho-S365 should be increased compared to wild-type BRAF, whereas S729 phosphorylation should be the same as wild-type.

Response: As kindly instructed, in **Supplementary Fig. 9p**, we found that similar to ubiquitination-deficient 5KR-BRAF (**Fig. 7b** and **Supplementary Fig. 10d-e**), ITCH binding-deficient BRAF mutant (S675A/P676A+P751A/P754, termed PA1/PA2) also displayed higher S365 phosphorylation in cells, while the S729 phosphorylation remain largely the same. In addition, we found that adding the S365A mutation to PA1/PA2-BRAF restored the activity of PA1/PA2-BRAF to phosphorylate MEK in 293 cells (**Supplementary Fig. 9o**). Furthermore, analogous to 5KR-BRAF, we found that PA1/PA2-BRAF could not be efficiently ubiquitinated by ITCH *in vitro* and thus still bound to 14-3-3 even after E1/E2/E3/UB incubation (**Supplementary Fig. 9f**).

6. The *in vivo* experiments in Fig 7 would be more relevant if knockdown and overexpression were inducible, so that the authors could study the effect of overexpression of 5KR-BRAF in established tumors.

Response: We thank the reviewer for this insightful comment. As kindly instructed, we generated WM1346 melanoma cell lines expressing a doxycycline-inducible EZ-Tet-pLKO-sh*BRAF* construct and one of the doxycycline-inducible pTRIPZ-RFP (as a negative control), -WT-BRAF or -5KR-BRAF constructs. As shown in the **Supplementary Fig. 10b**, doxycycline treatment led to efficient BRAF knockdown which turned off pMEK/pERK in cells expressing EZ-Tet-pLKO-sh*BRAF* and pTRIPZ-RFP. On the other hand, cells expressing EZ-Tet-pLKO-sh*BRAF* and pTRIPZ-WT-BRAF displayed doxycycline-responsive BRAF expression and MEK/ERK activation. Importantly, doxycycline treatment failed to maintain MEK/ERK activity in pTRIPZ-5KR-BRAF-expressing cells (**Supplementary Fig. 10b**). Next we compared the clonogenic survival among these cells, as shown in **Supplementary Fig. 11e-f**, doxycycline-induced expression of WT-BRAF displayed a higher cell growth compared to cells expressing 5KR-BRAF. Moreover, these cell lines were inoculated into the nude mice to examine their tumor growth followed by doxycycline treatment. Notably, doxycycline administration to mice with EZ-Tet-pLKO-sh*BRAF*/pTRIPZ-WT-BRAF tumor displayed the fastest tumor growth, while tumors bearing EZ-Tet-pLKO-sh*BRAF*/pTRIPZ-RFP ceased to grow after a few days of doxycycline treatment. Tumors with EZ-Tet-pLKO-sh*BRAF*/pTRIPZ-5KR-BRAF displayed a much slower growth compared to WT-BRAF expressing cells (**Supplementary Fig. 11i-j**). Examination of the tumor samples revealed that 5KR-BRAF-expressing cells exhibited a compromised pMEK/pERK levels compared to cells harboring WT-BRAF (**Supplementary Fig. 11l**). These results are largely consistent with our xenograft experiments using the pBabe-BRAF stable cell lines (**Fig. 7**), thus further support our major conclusion.

7. In Supp. Figure 1e, there are significantly higher levels of ubiquitination of BRAF(V600E) compared to WT. How does this relate to the overall model the authors propose?

Response: We thank the reviewer for pointing out that compared to WT-BRAF, the BRAF^{V600E} mutant was more susceptible to ITCH-mediated ubiquitinated as demonstrated in **Supplementary Fig. 1e**. Further, BRAF^{V600E} was poly-ubiquitinated even without the exogenous ITCH. To determine if ITCH is the major endogenous E3 ligase that promotes BRAF^{V600E} ubiquitination. In **Supplementary Fig. 1h**, we demonstrated that depletion of endogenous ITCH abolished BRAF^{V600E} ubiquitination, suggesting that BRAF^{V600E} is also an ITCH ubiquitin substrate. BRAF^{V600E} typically exhibits a more open and active conformation compared to its WT counterpart, therefore it might be more accessible to ITCH and exposes more lysine residues for ubiquitination. However, since the constitutively active V600E-BRAF mutant escapes most of the known inhibitory regulations (*Brummer T et al. (2006). Oncogene. 25, 6262–6276*), its ubiquitination might not significantly modulate its activity (**Supplementary Fig. 8a**). In support of this notion, our results demonstrated that although depletion of ITCH abolished BRAF ubiquitination in BRAF^{V600E}-expressing 1205Lu melanoma cells (**Supplementary Fig. 4g**), pMEK and pERK levels in 1205Lu cells were not affected (**Supplementary Fig. 4h**). These results suggest that ITCH-mediated BRAF ubiquitination controls the activation of WT-BRAF rather than the V600E-BRAF mutant. On the other hand, ITCH-mediated ubiquitination of BRAF^{V600E} might play a role in BRAF^{V600E} inhibition resistance, a direction worth of future investigation. In light of the reviewer's comment, we have also added a paragraph in the Discussion section of the revised manuscript to speculate possible roles of BRAF^{V600E} ubiquitination in acquired BRAF^{V600E} inhibition resistance.

REVIEWERS' COMMENTS:

Reviewer #1 (Remarks to the Author):

The authors have made significant efforts to address the raised concerns. Given the enormous biochemical and molecular data provided, it is considered to be sufficient enough for a publication.

Reviewer #2 (Remarks to the Author):

All comments and concerns have been successfully addressed by the authors.

Point-by-Point Responses to Reviewers' Critiques (NCOMMS-18-20081A)

REVIEWERS' COMMENTS

Reviewer #1 (Remarks to the Author):

The authors have made significant efforts to address the raised concerns. Given the enormous biochemical and molecular data provided, it is considered to be sufficient enough for a publication.

Reviewer #2 (Remarks to the Author):

All comments and concerns have been successfully addressed by the authors.

Response: Once again, we sincerely appreciate the insightful comments from both reviewers that have been very helpful in guiding us along our revision.